# EG4D: Explicit Generation of 4D Object without Score Distillation

**Qi Sun**[1,2,*]  **Zhiyang Guo**[1,*]  **Ziyu Wan**[2]  **Jing Nathan Yan**[3]  **Shengming Yin**[1]
**Wengang Zhou**[1,†]  **Jing Liao**[2,†]  **Houqiang Li**[1]
[1]University of Science and Technology of China
[2]City University of Hong Kong  [3]Cornell University

## Abstract

In recent years, the increasing demand for dynamic 3D assets in design and gaming applications has given rise to powerful generative pipelines capable of synthesizing high-quality 4D objects. Previous methods generally rely on score distillation sampling (SDS) algorithm to infer the unseen views and motion of 4D objects, thus leading to unsatisfactory results with defects like over-saturation and Janus problem. Therefore, inspired by recent progress of video diffusion models, we propose to optimize a 4D representation by explicitly generating multi-view videos from one input image. However, it is far from trivial to handle practical challenges faced by such a pipeline, including dramatic temporal inconsistency, inter-frame geometry and texture diversity, and semantic defects brought by video generation results. To address these issues, we propose EG4D, a novel multi-stage framework that generates high-quality and consistent 4D assets without score distillation. Specifically, collaborative techniques and solutions are developed, including an attention injection strategy to synthesize temporal-consistent multi-view videos, a robust and efficient dynamic reconstruction method based on Gaussian Splatting, and a refinement stage with diffusion prior for semantic restoration. The qualitative and quantitative evaluations demonstrate that our framework outperforms the baselines in generation quality by a considerable margin.

## 1 Introduction

Recent years have seen a surge in the development of generative models capable of producing intelligible text (Brown et al., 2020; Touvron et al., 2023; OpenAI, 2023), photo-realistic images (Ramesh et al., 2021; Rombach et al., 2022a;a; Sauer et al., 2023a), video sequences (Skorokhodov et al., 2022; Bahmani et al., 2023; Singer et al., 2023a), 3D (Chan et al., 2022; Poole et al., 2023; Lin et al., 2023; Tang et al., 2024b) and 4D (dynamic 3D) assets (Ren et al., 2023; Jiang et al., 2024b; Singer et al., 2023b). Particularly with 4D assets, manual creation is a laborious task that requires considerable expertise from highly skilled designers. Systems capable of automatically generating realistic and diverse 4D content could greatly streamline the workflows of artists and designers, potentially unlocking new realms of creativity through "generative art" (Bailey, 2020).

Due to the scarcity of open-sourced annotated multi-view dynamic data, previous works (Yin et al., 2023b; Xu et al., 2024; Bahmani et al., 2024a;b; Ren et al., 2023; Zhao et al., 2023; Jiang et al., 2024b; Singer et al., 2023b; Zheng et al., 2024; Ling et al., 2024) rely on the score distillation sampling (SDS) (Poole et al., 2023) or its variants from pre-trained 2D diffusion models to distill information about unseen views and motion of objects. Despite the impressive performance, their rendering results still suffer from highly saturated texture (Wang et al., 2023) and multi-face geometry (Janus problem) (Armandpour et al., 2023), thus leading to less photo-realistic generations.

Motivated by recent progress in video diffusion models (Voleti et al., 2024; Blattmann et al., 2023a; Brooks et al., 2024), we propose a novel multi-stage framework, **EG4D**, for **E**xplicitly **G**enerating **4D** videos and then reconstructing 4D assets from them. EG4D goes beyond simply adapting video

---

*Equal contributions; † Correspondence authors: Jing Liao and Wengang Zhou;
Code available: github.com/jasongzy/EG4D

generation results, as the synthesized frames inevitably suffer from temporal inconsistency and limited visual quality. More specifically, in the vanilla "frame-by-frame" reconstruction, the independence and diversity of multi-view diffusion will cause appearance inconsistency across different timestamps, particularly in unseen views.

To address these challenges, we first design an attention injection mechanism, allowing each multi-view diffusion inference to perceive temporal information through cross-frame latent exponential moving average (EMA). This training-free strategy effectively alleviates the inconsistency issue at the video level and ensures high-quality training samples for optimizing the following 4D representation. In the next stage of 4D reconstruction, we choose 4D Gaussian Splatting (4D-GS) (Wu et al., 2024a) as our representation to take advantage of its efficient training and rendering capability.

Moreover, existing GS-based dynamic reconstruction methods (Wu et al., 2024a; Yang et al., 2023; Huang et al., 2024b) commonly assume that appearance variations between different timestamps are caused by the geometric deformation of Gaussian splats. However, this assumption does not hold since unwanted color variations of texture details still exist in our synthesized images produced by video diffusions, even with the proposed attention injection strategy. We manage to disentangle such detailed texture inconsistencies from desired geometric deformation by introducing an extra color transformation network, enabling texture-consistent 4D rendering. Furthermore, we leverage image-to-image diffusion models to refine the rendered images and fine-tune our 4D representation, achieving better generation quality.

The qualitative results, quantitative evaluations and user preferences validate that our EG4D outperforms SDS-based baselines by a large margin, producing 4D content with realistic 3D appearance, high image fidelity, and fine temporal consistency. Extensive ablation studies also showcase our effective solutions to the challenges in reconstructing 4D representation with synthesized videos.

## 2 RELATED WORKS

In this section, we present the recent progress of video diffusion models and 4D generation. More discussion on related works can be found in Appendix A.

**Video diffusion models.** Diffusion models (Ho et al., 2020), characterized by their superior generative capabilities, have become dominant in the field of video generation (Ho et al., 2022b; Singer et al., 2022b; Ho et al., 2022a; Blattmann et al., 2023a; Yin et al., 2023a). Among them, VDM (Ho et al., 2022b) replaces the typical 2D U-Net for modeling images with a 3D U-Net. Make-A-Video (Singer et al., 2022b) successfully extends a diffusion-based T2I model to T2V without text-video pairs. Text2Video-Zero (Khachatryan et al., 2023) achieve zero-shot text-to-video generation using only a pre-trained text-to-image diffusion model without any further fine-tuning or optimization. Following Latent Diffusion Models (Rombach et al., 2022b), Video-LDM (Blattmann et al., 2023b) and AnimateDiff (Guo et al., 2024a) introduce additional temporal layers designed to model the temporal consistency. Stable Video Diffusion (Blattmann et al., 2023a), trained on well-curated high quality video dataset, presents robust text-to-video and image-to-video generation capabilities across various domains. Recently, SV3D (Voleti et al., 2024) adapts image-to-video generation for novel view synthesis by leveraging the generalization and multi-view consistency of the video models. Different from these works, we aim to explicitly generate 4D videos with both temporal and multi-view consistency using two orthogonal video diffusion models.

**4D generation.** Following the line of text-to-3D synthesis (Poole et al., 2023; Wang et al., 2023; Wan et al., 2024), one line of research explores the text-conditioned 4D generation (Yin et al., 2023b; Cai et al., 2023; Xu et al., 2024; Bahmani et al., 2024a; Zheng et al., 2024; Bahmani et al., 2024b; Singer et al., 2023b; Zheng et al., 2024; Ling et al., 2024). They use score distillation sampling (SDS) (Poole et al., 2023) to optimize the 4D representations, like KPlanes (Fridovich-Keil et al., 2023), Hexplanes (Cao & Johnson, 2023) and Deformable Gaussians (Yang et al., 2023). MAV3D (Singer et al., 2023b) employs temporal SDS to transfer the motion from text-to-video diffusions (Singer et al., 2022a)) to a dynamic NeRF. AYG (Ling et al., 2024) explores compositional 4D generation with 3D Gaussian Splatting. Inspired by recent advancement in image-to-3D models (Liu et al., 2023b;a; Shi et al., 2024), several works (Zhao et al., 2023; Ren et al., 2023; Jiang et al., 2024b; Pan et al., 2024) explore the field of image/video-conditioned 4D generation. Animate124 (Zhao et al., 2023) pioneers on this task in a coarse-to-fine fashion: it first optimizes

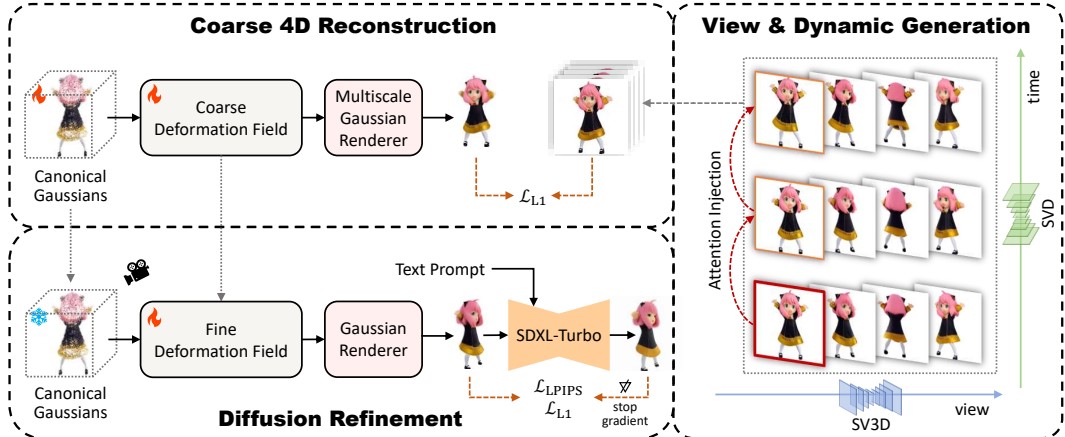

Figure 1: **Framework of EG4D.** In video generation (right, Sec. 4.1), we use SVD to produce dynamic frames, and then use SV3D equipped with attention injection to generate temporal-consistent multi-view images. In coarse 4D reconstruction (left top, Sec. 4.2), we optimize the 4D Gaussian Splatting with additional color affine transformation with the annotated multi-view images produced by Stage I. In diffusion refinement (left bottom, Sec. 4.3), we freeze the canonical Gaussians and fine-tune the temporal deformation network with images refined by image-to-image diffusion model.

deformation with multi-view diffusions, then corrects the details with ControlNet (Zhang et al., 2023). DreamGaussian4D (Ren et al., 2023) adopts explicit modeling of spatial transformations in Gaussian Splatting, achieving minute-level generation. L4GM (Ren et al., 2024) trains a large-scale feed-forward network for Gaussian sequences to extend LGM (Tang et al., 2024a) in the temporal dimension without optimization. Recently, some concurrent works (Xie et al., 2024; Zeng et al., 2024; Liang et al., 2024; Jiang et al., 2024a) also try to exploit the video diffusion models for 4D content generation, by jointly optimize the temporal and view dimensions to train a 4D-aware video diffusion model. Although achieving appearance consistency, this approach entangles motion and multi-view generating processes, and typically requires limited 4D datasets and considerable computing resources for the training. Although score distillation algorithm can infer motion and unseen views from 2D diffusion, it suffers from imperfections like over-saturation and Janus problem. Our framework gets around the above problems by explicitly generating temporal-consistent multi-views of dynamic objects in a training-free manner, and then uses them to reconstruct 4D representations.

## 3 PRELIMINARIES

**Video diffusion.** In this work, we use two different video diffusion models: Stable Video Diffusion (Blattmann et al., 2023a) (SVD) and SV3D (Voleti et al., 2024). SVD generates a sequence of video frames $\{I_t | t \in \{0, \cdots, T\}\}$ conditioned on an initial image $I_0$ or text prompt. SV3D is a pose-conditioned image-to-multiviews model that takes a reference image $I_0$ and a series of camera poses $\{c_p | p \in \{1, \cdots, N\}\}$, producing a sequence of video frames $\{I_p | p \in \{1, \cdots, N\}\}$ corresponding to the specified pose (camera parameters) sequence. Both SVD and SV3D adopt similar video diffusion architecture (Ling et al., 2024) with spatial and temporal attention layers.

**3D Gaussian Splatting.** 3DGS (Kerbl et al., 2023) is an explicit representation using millions of 3D Gaussians to model a scene. Each Gaussian is characterized by a set of learnable parameters as follows: **1)** 3D center; **2)** 3D rotation; **3)** 3D size (scaling factor); **4)** view-dependent RGB color represented by $k$-DoF spherical harmonics coefficients: $\boldsymbol{h} \in \mathbb{R}^{3(k+1)^2} \to \boldsymbol{c} \in \mathbb{R}^3$; **5)** opacity. Here a color decoder $\Phi^{sh}$ is used to turn the spherical harmonics coefficients $\boldsymbol{h}$ and the view direction $\boldsymbol{\gamma}$ into an actual RGB color $\boldsymbol{c}$. For a position in the scene, each Gaussian makes its contribution at that coordinate according to the standard Gaussian function weighted by its opacity. The differentiable rendering of 3DGS applies the splatting techniques (Kerbl et al., 2023). For a certain pixel, the point-based rendering computes its color by evaluating the blending of depth-ordered points overlapping that pixel via the volume rendering equation (Max, 1995). The optimization of Gaussian parameters is then supervised by the reconstruction loss (difference between rendered and ground-truth images).

# 4 4D OBJECT GENERATION

Given an object image, we want to generate the 4D representation of it, enabling free-view dynamic rendering. To this end, we introduce a multi-stage framework (generation-reconstruction-refinement) for 4D object generation, as illustrated in Figure 1. Network details can be found in Appendix B.

## 4.1 STAGE I: VIEW AND DYNAMIC GENERATION WITH VIDEO DIFFUSIONS

In this stage, we employ two orthogonal video diffusion models to generate samples for the later 4D representation optimization. Given a reference image, we use SVD (Blattmann et al., 2023a) to generate a sequence of video frames $\{I_t | t \in \{0, \cdots, T\}\}$, where $t$ is the timestamp. Next, we utilize SV3D (Voleti et al., 2024) to generate multi-view images $\{I_{t,p} | p \in \{1, \cdots, N\}\}$ with a predefined camera pose sequence for each frame $I_t$. However, vanilla "frame-by-frame" reconstruction causes significant *temporal differences* due to the diverse nature of SV3D inferences for those frames. Hence, we hope to exploit temporal context to guide the otherwise independent generating process, thereby obtaining results that are as temporally consistent as possible. To this end, we introduce the training-free *attention injection* strategy during our SV3D inference.

Specifically, in each self-attention module of the spatial layers of a diffusion UNet, we simultaneously consider the visual information from the current reference frame and the frames at previous timestamps, and implement the attention injection by *spatial KV latent blending* formulated as

$$z_t \leftarrow \alpha z_t^* + (1 - \alpha) z_{t-1}, \tag{1}$$

$$Q = W^q z_t^*, K = W^k z_t, V = W^v z_t, \tag{2}$$

$$\text{Attention}(Q, K, V) = \text{Softmax}(\frac{QK^T}{\sqrt{d_k}} V), \tag{3}$$

where $z_t$ is the exponential moving average (EMA) of the current multi-view latent $z_t^*$ and the one from the previous timestamp $z_{t-1}$, with the blending weight $\alpha$. $d_k$ is the key dimension.

## 4.2 STAGE II: COARSE RECONSTRUCTION WITH GAUSSIAN SPLATTING

With the synthesized multi-view images $\{I_{t,p} | t \in \{0, \cdots, T\}, p \in \{1, \cdots, N\}\}$ of the dynamic object, we optimize a 4D representation of it to enable free-viewpoint rendering. It is worth noting that in this stage, our objective is *not* simply reconstructing an object according to multi-view observations. Although the design in Sec. 4.1 significantly alleviates the temporal inconsistency problem, those synthesized "ground-truth" images still suffer from varying degrees of inconsistency in color details. Therefore, we propose to optimize a 4D representation based on 3D Gaussian Splatting (Kerbl et al., 2023) with additional insights into the robustness against texture inconsistencies and semantic defects.

**Canonical Gaussians & deformation field.** Considering both performance and efficiency, we build our 4D representation upon 4D Gaussian Splatting (4D-GS) (Wu et al., 2024a). 4D-GS utilizes a deformation field to predict each Gaussian's geometric offsets at a given timestamp relative to a mean canonical state. This deformation field is composed of a multi-resolution HexPlane (Cao & Johnson, 2023) and MLP-based decoders. For each Gaussian at a certain timestamp, the model queries the Hexplane with a 4D coordinate ($x$-$y$-$z$-$t$) and decodes the obtained feature $f_t$ into the position, rotation, and scaling deformation values. The entire dynamic scene is then jointly reconstructed by optimizing both canonical Gaussians and the deformation field, enabling implicit global interactions of visual information.

**Color transformation against texture inconsistency.** While vanilla 4D-GS is theoretically able to model temporal inconsistencies through per-frame geometric deformation of Gaussians, it is hard to optimize and leads to significant redundancy in Gaussian quantity (Guo et al., 2024b). Even if all the inconsistencies are faithfully reconstructed, these unnatural variations in texture details across time will result in significant degradation of visual performance. To address this problem,

we want to disentangle such detailed texture inconsistencies from geometric deformation. Those temporal differences can still be modeled as per-timestamp states, while one of them can be manually selected to dominate the final temporal-consistent rendering. We choose a simple but effective way that performs time-specific color transformation. Formally, a new color decoder denoted by $\Phi^c$ is introduced as follows:

$$\boldsymbol{c} = \Phi^c(\boldsymbol{h}, \boldsymbol{\gamma}) = \boldsymbol{W}_t^c \Phi^{sh}(\boldsymbol{h}, \boldsymbol{\gamma}) + \boldsymbol{b}_t^c, \tag{4}$$

$$\boldsymbol{W}_t^c, \boldsymbol{b}_t^c = \text{MLP}(\boldsymbol{f}_t), \tag{5}$$

where $\boldsymbol{h}$ is the spherical harmonics coefficients of Gaussians, $\boldsymbol{\gamma}$ is the view direction, and $\Phi^{sh}$ is the spherical harmonic decoder. $\boldsymbol{W}_t^c$ and $\boldsymbol{b}_t^c$ are weights and bias predicted by an extra MLP-based color head from per-Gaussian time-specific feature $\boldsymbol{f}_t$ from the HexPlane. Such kind of affine transformation is competent in modeling texture inconsistencies caused by ambient occlusion and other factors (Li et al., 2022; Darmon et al., 2024). During 4D rendering at test time, we take one of the timestamps, *e.g.*, the first frame, as the reference time and use the corresponding feature $\boldsymbol{f}_0$ to get the Gaussian colors, thereby rendering texture-consistent 4D assets.

**Multiscale rendering augmentation.** Generally, for a reconstruction task, supervision with high-resolution ground-truth images can provide more information about high-frequency details and benefit the rendering quality. However, in our task, those synthesized images often have high-frequency noises at specific views or timestamps. Training with them leads to meaningless view- and frame-overfitting and adds more burden to later refinement. To address this issue, we propose a multiscale augmentation strategy. During optimization, we randomly downsample the ground-truth images within a reasonable ratio range. The rendering parameters of the Gaussian rasterizer are modified accordingly, enabling multiscale supervision with the reconstruction loss.

### 4.3 STAGE III: REFINEMENT WITH DIFFUSION PRIORS

Videos produced by diffusion models often suffer from semantic defects (Figure 2 left) and motion blur. Fortunately, image-to-image diffusion models provide a strong prior to refine the semantic details while preserving object identity and style. We leverage these diffusion-refined images (Figure 2 right) to fine-tune our 4D representation further. Specifically, we first render an image $I_{t,p}$ at the timestamp $t$ and camera pose $p$. Then we encode the image $I_{t,p}$ into a VAE latent $w$, add noise to the latent, and feed it into the diffusion UNet for denoising. Finally, the refined image $\hat{I}$

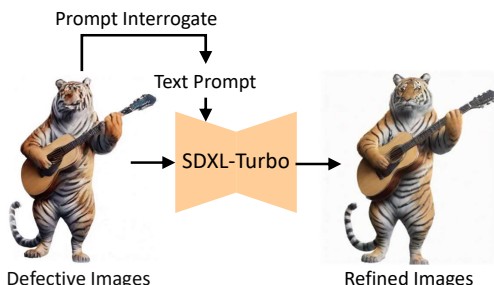

Figure 2: **Illustration of diffusion refinement.**

is decoded from the denoised latent $\hat{w}$. Additionally, to account for per-view quality variations, we introduce a pre-defined view-dependent weight $f(p)$ to the reconstruction loss. Empirically, we select a sine scheduler for pose-dependent weight, formulated as $f(p) = \sin(\pi \cdot \text{d}(x_p, x_0))$, where $x_0$ is the camera center of the first frame and $\text{d}(\cdot, \cdot)$ is the normalized L2 distance. In total, the diffusion refinement loss $\mathcal{L}_{\text{ref}}$ is formulated as

$$\mathcal{L}_{\text{ref}} = f(p) \cdot (\mathcal{L}_{\text{L1}}(I_{t,p}, \hat{I}) + \lambda \cdot \mathcal{L}_{\text{LPIPS}}(I_{t,p}, \hat{I})), \tag{6}$$

where $\mathcal{L}_{\text{LPIPS}}(\cdot, \cdot)$ is the perceptual loss and $\mathcal{L}_{\text{L1}}(\cdot, \cdot)$ is the pixel-wise $\text{L}_1$ loss. To preserve the coarse geometry and texture from Stage II, we use this loss to fine-tune the coarse deformation field while keeping canonical Gaussians frozen. To avoid error accumulation and unstable supervision, we conduct one-pass refinement: for each view/timestamp, the rendered image at the first iteration are used as the input of diffusion, and the refinement output is shared with all the iterations later.

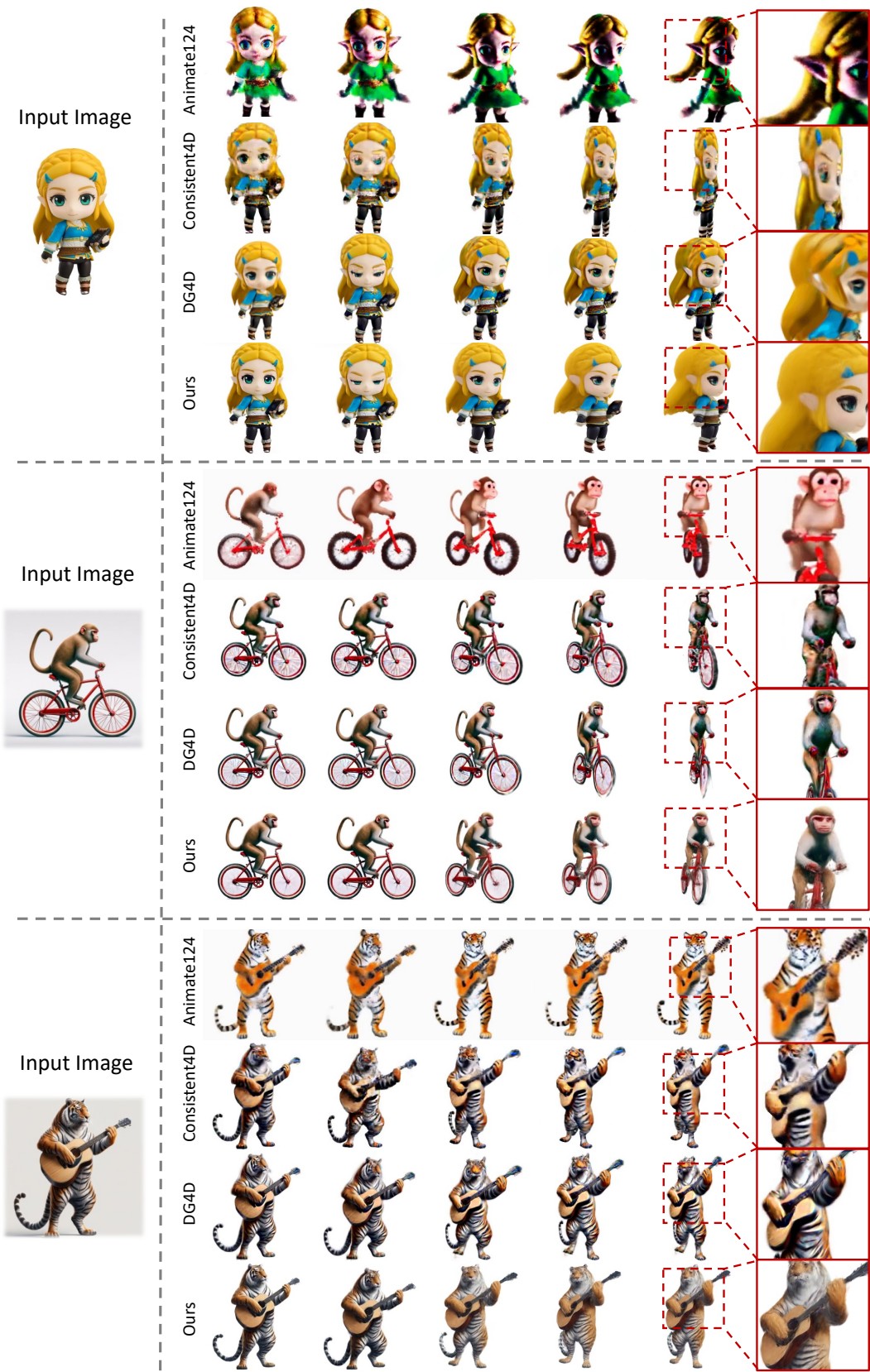

Figure 3: Comparison with Animate124 (Zhao et al., 2023), Consistent4D (Jiang et al., 2024b), and DreamGaussian4D (DG4D) (Ren et al., 2023) in three cases `zelda`, `monkey-bike` and `tiger-guitar` (better zoom in). The first two columns show the animation results in the same view, and the 3-5 columns demonstrate three other views.

Input Image                 View 1                 View 2

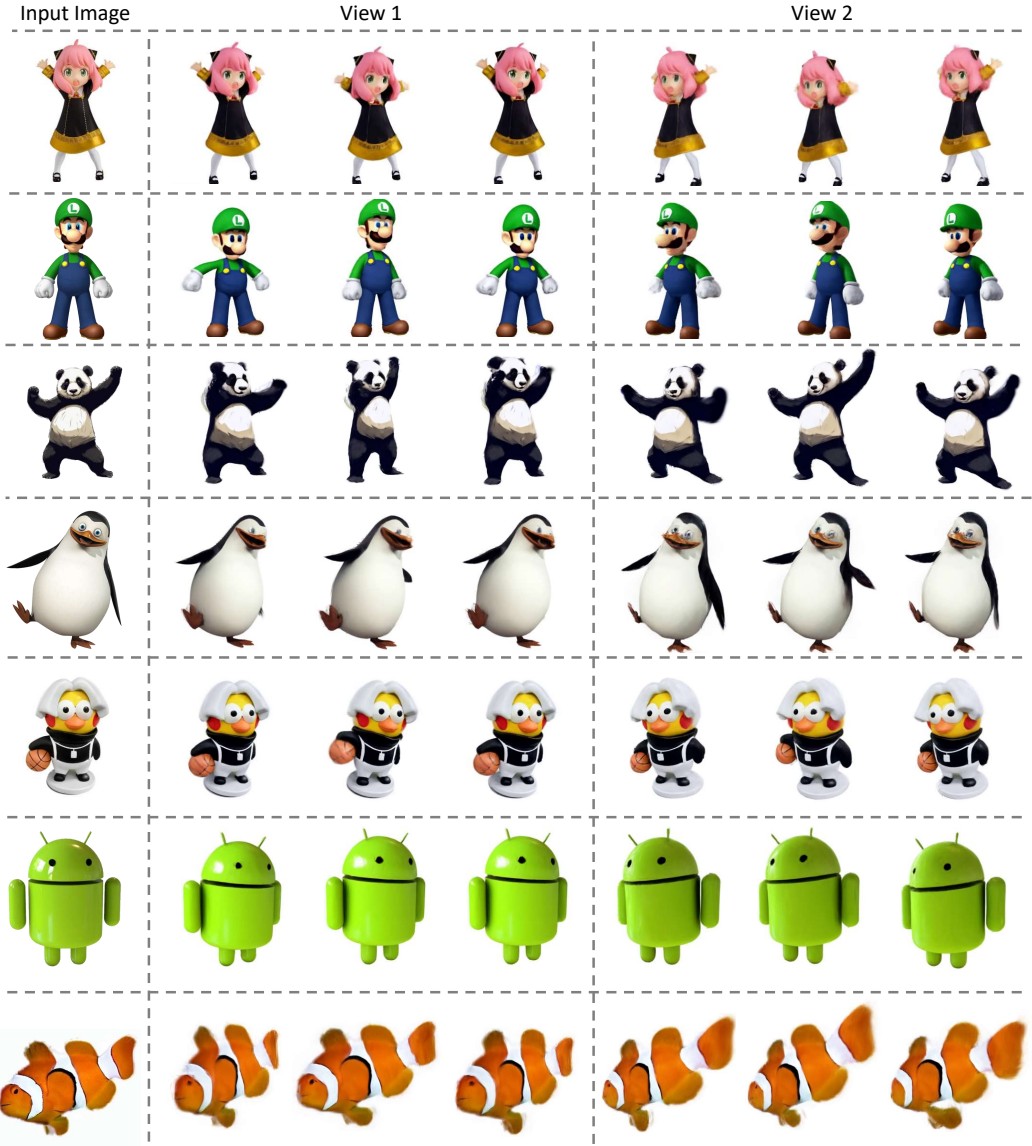

Figure 4: **Qualitative results of our generated 4D objects.** We present three consecutive frames rendered from our 4D model from two different views.

## 5 EXPERIMENTS

### 5.1 EXPERIMENTAL SETTINGS

**Implementation details.** In Stage I, we use SVD-img2vid-xl (Blattmann et al., 2023a) to generate 25-frame videos. For multi-view generation, we employ SV3D$^p$ conditioned on a camera pose sequence, *i.e.*, 21 azimuth angles (360° evenly divided) and a fixed 0° elevation. All images are set to a resolution of 576×576. In Stage III, we use SDXL-Turbo (Sauer et al., 2023b) with small strength (0.167) to provide the diffusion prior. More reproduction details are included in Appendix C.1.

**Evaluation metrics.** Following previous methods (Ren et al., 2023; Zhao et al., 2023), we use CLIP-I score that measures the cosine similarity of CLIP (Radford et al., 2021) embedding of the given image and the rendered views. We use novel view synthesis metrics like PSNR, LPIPS, SSIM between the given image and rendered results to measure the image quality. We also use FVD (Unterthiner et al., 2018) between reference video and 4D rendered video to assess the generated 4D quality. In consideration of quantifying the temporal consistency, we adopt a recent proposed metric, CD-FVD (Ge et al., 2024), which prefers temporal consistency instead of the per-frame quality.

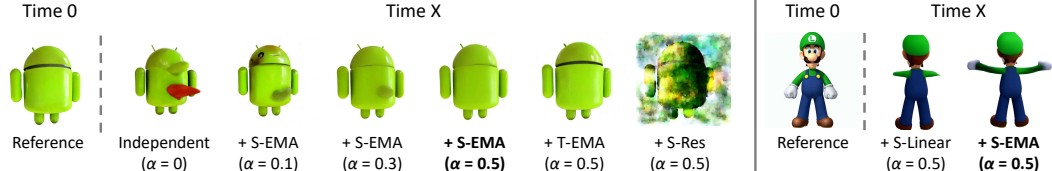

Figure 5: **Ablation on attention injection**. Video generation results are shown with two cases at time 0 and a timestamp X afterward. "S-" and "T-" stand for operations in spatial and temporal attention layers of SV3D, respectively. "EMA" denotes the proposed KV latent blending with Exponential Moving Average. "Linear" denotes KV blending with only the first frame. "Res" denotes injection on residual connection instead of KV. $\alpha$ is the blending weight. Different degrees of temporal inconsistency can be observed in all settings except ours (S-EMA, $\alpha = 0.5$).

Table 1: **User study on image-to-4D methods.** Each number represents the percentage of user preference. Error bars correspond to the 95.6% confidence interval. **Bold** denotes the best result.

| Method | Overall Quality | Ref. View Alignment | 3D Appearance | Motion Realism | Motion Range |
|---|---|---|---|---|---|
| Animate124 (Zhao et al., 2023) | 1.10 ±1.24 | 2.24 ±1.77 | 1.65 ±1.52 | 2.19 ±1.75 | 5.39 ±2.70 |
| Consistent4D (Jiang et al., 2024b) | 3.88 ±2.31 | 5.00 ±2.60 | 3.99 ±2.34 | 5.66 ±2.76 | 8.67 ±3.36 |
| DreamGaussian4D (Ren et al., 2023) | 11.27 ±3.78 | 12.17 ±2.91 | 10.29 ±3.63 | 15.42 ±4.32 | 39.96 ±5.83 |
| EG4D (Ours) | **83.75** ±4.41 | **80.59** ±4.73 | **84.07** ±4.37 | **76.73** ±5.05 | **45.98** ±5.93 |

It uses the features extracted by self-supervised video representation learning model, instead of pre-trained I3D (Carreira & Zisserman, 2017) feature, to mitigate this content bias of FVD to a large extent. We also conduct a user preference study to evaluate the 3D appearance, view alignment, motion realism, motion range, and overall quality. Details on user study can be found in Appendix C.3.

**Baselines.** We compare our results with the state-of-the-art open-sourced image-to-4D methods: Animate124 (Zhao et al., 2023) and DreamGaussian4D (Ren et al., 2023). We also compare with the state-of-the-art video-to-4D method Efficient4D (Pan et al., 2024) and Consistent4D (Jiang et al., 2024b). For a fair comparison, we feed the SVD-generated videos (same as ours) to video-based methods/DreamGaussian4D for direct video-to-4D generation.

## 5.2 RESULTS

**Qualitative results.** Figure 3 demonstrates three cases for comparison between our EG4D and the baselines (Ren et al., 2023; Zhao et al., 2023; Jiang et al., 2024b). Our generated results present better image-4D alignment and more realistic 3D appearance, especially in facial details. Animate124 can not generate image-aligned 4D models because of its strong text guidance. Consistent4D and DreamGaussian4D produce models with over-saturated and non-realistic appearance (especially in face) due to the inherent limitation of score distillation algorithm. Figure 4 shows detailed results produced by EG4D. For each case, we present three temporal-continuous rendered frames from two views. *Note that our framework is not limited to SVD, which can hardly produce large motions. Our approach can seamlessly incorporate advanced I2V models such as AnimateAnyone (Hu, 2024), enabling larger and controllable motion synthesis..*

**Quantitative results & User study.** Table 2 shows that our method has the highest CLIP-I score, which means the rendered images are more semantically similar to the reference image. In the novel view synthesis metrics (PSNR, SSIM, LPIPS),

Table 2: **Quantitative results** in Animate124 benchmark. **Bold** denotes the best result.

| Method | CLIP-I ↑ | PSNR ↑ | SSIM ↑ | LPIPS ↓ | FVD↓ | CD-FVD↓ |
|---|---|---|---|---|---|---|
| Animate124 (Zhao et al., 2023) | 0.8544 | 9.972 | 0.643 | 0.361 | 1904.1 | 2435.9 |
| Consistent4D (Jiang et al., 2024b) | 0.9214 | 14.17 | 0.767 | 0.225 | 736.02 | 1005.8 |
| Efficient4D (Pan et al., 2024) | 0.9358 | 20.38 | 0.822 | 0.179 | 753.11 | 891.33 |
| DreamGaussian4D (Ren et al., 2023) | 0.9227 | 9.039 | 0.637 | 0.512 | 801.10 | 816.23 |
| EG4D (Ours) | **0.9535** | **23.28** | **0.904** | **0.173** | **142.34** | **459.10** |

our results surpass the baselines by a very large margin, which highlight the advantage of pixel-wise optimization of our 4D model over the score distillation. The substantial improvement of FVD and CD-FVD indicate that our framework produces 4D object with the best per-frame quality and temporal consistency. User study (Table 1) shows that the recipients are overwhelmingly inclined towards

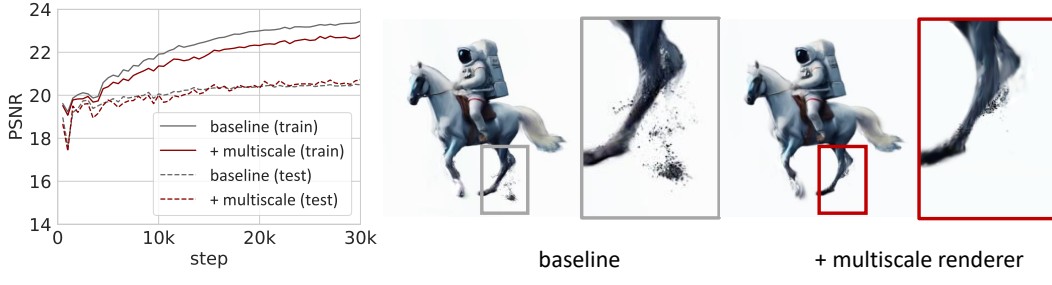

(a) Reconstruction quality comparison.  (b) Visual details illustration.

Figure 7: **Effects of multiscale renderer.** **(a)** demonstrates the training (solid line) / test (dashed line) curve before (dark gray) and after (dark red) adding the multiscale renderer. Multiscale rendering avoids the meaningless overfitting of our model (lower training PSNR, but comparative or even higher test PSNR). **(b)** shows one viewpoint of rendering for case `astronaut-horse`. The multiscale render effectively prevents the model from overfitting to noise introduced in video diffusions.

the 4D results generated by our framework. Almost 80% of the participants think our method is superior in overall quality, reference view consistency, 3D appearance, and motion realism. Meanwhile, our motion range is on par with the strongest baseline, which is further discussed in Sec. 6.

### 5.3 ABLATION STUDIES

**Attention injection.** In Figure 5, we explore the effect of attention injection by generating videos (Stage I) with different blending weight $\alpha$ and replacing our spatial KV latent blending with three variants: **1)** *T-EMA*: similar KV blending is adopted but in temporal attention layers of SV3D, *i.e.*, one frame is blended with all views of the reference timestamp, which results in almost identical (static) results. **2)** *S-Res*: the residual term (skip connection) instead of KV latent is blended, which leads to collapse results. **3)** *S-Linear*: KV blending is used but only with the first frame. Without EMA, the diffusion model shows degraded generating capability for large motions departing from the reference frame (`luigi` in Figure 5 right). Moreover, we observe that the temporal consistency is highly sensitive to blending weight $\alpha$. For comparison, without any attention injection strategy ($\alpha = 0$), views of different timestamps are generated independently, leading to dramatic temporal inconsistency in the back view of `android` (Figure 5 left). Our proposed spatial KV blending with EMA effectively improves the consistency when $\alpha$ is increased to 0.5. Please refer to Appendix D for the dynamic attenuation phenomenon when $\alpha > 0.5$.

**Color transformation.** Figure 6 shows the effectiveness of our proposed color transformation in Stage II. Dynamic 3DGS typically models all the inter-frame

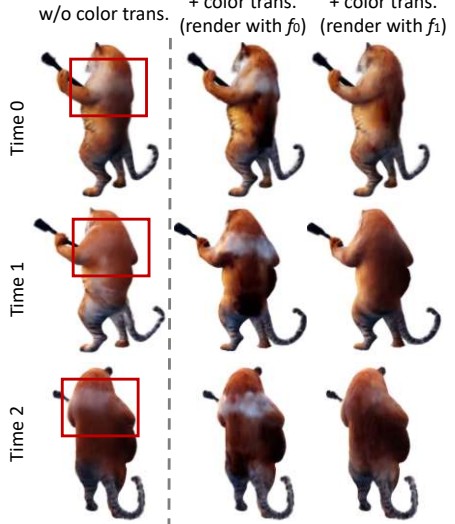

Figure 6: **Effects of color transformation.** Our color affine transformation effectively disentangles the texture variation at different timestamps, enabling the rendering of color-consistent dynamics with arbitrary time-specific feature $\boldsymbol{f}_t$.

texture diversity as part of time-specific deformation. With color affine transformation, we manage to disentangle unwanted color inconsistencies and render temporal-consistent texture details from whichever timestamp we select.

**Multiscale renderer.** Figure 7 shows the effectiveness of our multiscale renderer of Stage II. We show the training and test PSNR during optimization in the left panel. It can be observed that the multiscale renderer plays the role of a regularizer that effectively avoids model overfitting (lower training PSNR and similar test PSNR). The qualitative result in the right panel illustrates that this design avoids overfitting to the noise introduced by the video diffusions.

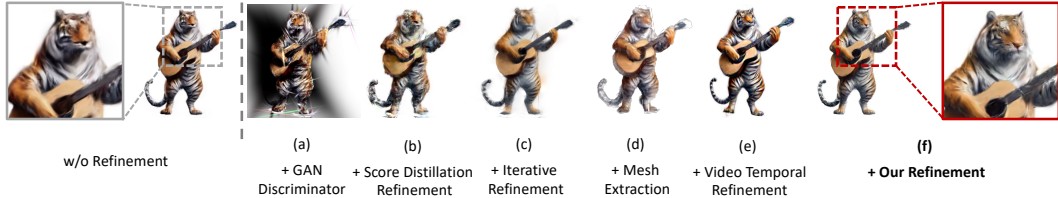

w/o Refinement

(a) + GAN Discriminator
(b) + Score Distillation Refinement
(c) + Iterative Refinement
(d) + Mesh Extraction
(e) + Video Temporal Refinement
**(f) + Our Refinement**

Figure 8: **Ablation on different refinement methods.** The leftmost column shows the image rendered by 4D model after the second stage optimization for the case `tiger-guitar`. Panels (a)-(f) demonstrate different refinement methods aimed at addressing the semantic defects. However, only the one-pass refinement (ours) successfully adds facial details while keeping the original structure.

**Refinement strategies.** Figure 8 illustrates ablations of refinement by comparing the visual details before and after applying various refinement techniques. **(a)** *Adversarial training*: many previous works (Chen et al., 2024; Roessle et al., 2023) leverage a GAN discriminator to optimize neural fields. However, we observe that although the discriminator loss converges quickly, the Gaussian points gradually diverge from the object surface, resulting in rendered images turning black after several iterations. **(b)** *SDS (score distillation refinement)*: SDS seeks a single mode for text-aligned 4D representation, leading to unsuccessful refinement. **(c)** *Iterative refinement*: InstructN2N (Haque et al., 2023) iteratively updates the supervised dataset (each image is refined for multiple times, different from our *one-pass refinement*) for 3D scene editing. In our task, the diverse outputs from diffusion model result in blurred 4D model under pixel-wise supervision. **(d)** *Textured mesh extraction* (Tang et al., 2024b): experiments show that meshes extracted from 3D Gaussians are not watertight and smooth (Tang et al., 2024b; Huang et al., 2024a), leading to incoherent appearance. **(e)** *Video temporal refinement* (Ren et al., 2023): SVD prior alone is insufficient for structure preservation and detail refinement. **(f)** *One-pass refinement* (Ours): in this way, the refined (supervised) images strike a balance between detail restoration and preservation of structural integrity and consistency. This approach introduces reasonable details in noisy or semantically defective regions. Temporal consistency is maintained despite frame-by-frame inference, as input images are similar (two renderings of the same object). The similarity of inputs and predefined text prompt ensure consistent noise directions and results in converging outputs in the image-to-image diffusion process (Haque et al., 2023). Quantitative ablation (Table 3) demonstrates that this stage preserves temporal consistency while significantly enhancing per-frame image quality.

Table 3: **Ablation study**

| Method | FVD↓ | CD-FVD↓ |
|---|---|---|
| w/o Stage III | 179.93 | **448.57** |
| w. Stage III | **142.34** | 459.10 |

## 6  CONCLUSION AND DISCUSSION

**Conclusion.** In this paper, we propose EG4D, a novel framework for 4D generation from a single image. This approach departs from previous score-distillation-based methodologies, promising not only intrinsic immunity against problems like over-saturation but also capabilities for consistent visual details and dynamics. We first equip the video diffusions with a training-free attention injection strategy to explicitly generate consistent dynamics and multi-views of the given object. Then a coarse-to-fine 4D optimization scheme is introduced to further address practical challenges in synthesized videos. Qualitative and quantitative results demonstrate that EG4D produces 4D objects with more realistic and higher-quality appearance and motion compared with the baselines.

**Limitations & Future work.** One limitation is that our framework can not generate high-dynamic motion due to the limited capability of the base image-to-video model (Blattmann et al., 2023a) and the consistency-motion trade-off in our attention injection strategy. Another problem lies in the multi-view diffusion model (Voleti et al., 2024), which currently struggles to apply precise camera pose conditioning, leading to unsatisfactory reconstruction. One solution for dynamics is to leverage more advanced video diffusions to generate high-quality and high-dynamic video frames. Future work could also incorporate adaptive camera pose techniques (Fu et al., 2024; Smith et al., 2024) in 4D reconstruction to further improve the robustness.

# 7 Acknowledgments

We thank all anonymous reviewers and area chairs for their valuable comments. The work described in this paper was substantially supported by a GRF grant from the Research Grants Council (RGC) of the Hong Kong Special Administrative Region, China [Project No. CityU 11208123].

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

# Appendix

In this appendix, we first present additional related works on 3D refinement (Appendix A). Then we provide detailed network specifications (Appendix B). Next, To ensure reproducibility and facilitate fair perceptual studies, we describe the experimental settings in detail (Appendix C). Finally, we include extended ablation studies (Appendix D) and additional results (Appendix E) to demonstrate the robustness and superiority of our methods across various settings.

## A    MORE RELATED WORKS

**3D refinement with generative priors.**    To deal with view-inconsistency and low quality problems, many works (Wu et al., 2024c; Roessle et al., 2023; Chen et al., 2024; Wu et al., 2024b; Haque et al., 2023; Vachha & Haque, 2024; Zhou & Tulsiani, 2023) take advantages from generative priors, *e.g.*, adversarial training (Goodfellow et al., 2014) and score distillation sampling (SDS) (Poole et al., 2023) to optimize the 3D representation. GANeRF (Roessle et al., 2023) refines the rendered images with an image-conditional generator and leverages the re-rendered image constraints to guide the NeRF optimization in the adversarial formulation. InstructNeRF2NeRF (Haque et al., 2023) uses the text-conditioned image generator, InstructPix2pix (Brooks et al., 2023), to edit the image rendered by pre-trained NeRF in an iterative manner and updates the underlying 3D representation with the edited images. ReconFusion (Wu et al., 2024b) uses the diffusion priors, Zero-123 (Liu et al., 2023b), as a drop-in regularizer to enhance the 3D reconstruction performance, especially for sparse-view scenarios. In contrast to directly optimizing the implicit representation, another line of researches (Tang et al., 2024b; Ren et al., 2023) first extracts the explicit textured mesh, and then refine the texture in UV-space with diffusion prior and differentiable rendering. In particular, DreamGaussian4D leverages SVD as image-to-video prior to enhance the texture temporal consistency. In our paper, in consideration of the artifacts generated in video diffusion, we extend the refinement techniques to the 4D representation.

## B    NETWORK DETAILS

In this section, we unpack the core network design in Figure 1.

**Attention injection.**    In Sec. 4.1, we exploit the attention injection strategy to alleviate the temporal difference between multi-view diffusion models. Figure 9 illustrates its network details: in each spatial attention layer, we replace the self-attention by simultaneously considering the current $z_t^*$ and previous visual information with EMA.

**Deformation field with color transformation.**    In Sec. 4.2, we use color affine transformation to model the temporal texture variation. Figure 10 shows the detailed architecture of that. We first query the time-specific feature $f_t$ from the learnable HexPlane (Cao & Johnson, 2023) with the canonical Gaussian positions $\bar{\mu}$. After that, the geometric deformations of Gaussian properties ($\mu$ location, $r$ rotation, and $s$ scale) are predicted with a lightweight decoder. Additionally, we use the affine color transformation to model the temporal texture variations. Finally, these deformed Gaussians are rendered into an image.

## C    ADDITIONAL EXPERIMENTAL SETTINGS

### C.1    OPTIMIZATION DETAILS

We report the optimization of 4D Gaussian splatting for the purpose of reproduction. Basically, we follow the training recipe from 4DGS (Wu et al., 2024a) in the coarse 4D reconstruction stage. In the semantic refinement stage (Stage III), we fine-tune 4DGS for 5k steps with Adam optimizer. The initial learning rate is set to 1e-4 with exponential decay. The weight $\lambda$ in diffusion refinement loss is set to 0.5. Our implementation is primarily based on the PyTorch framework and tested on a single NVIDIA RTX 3090 GPU.

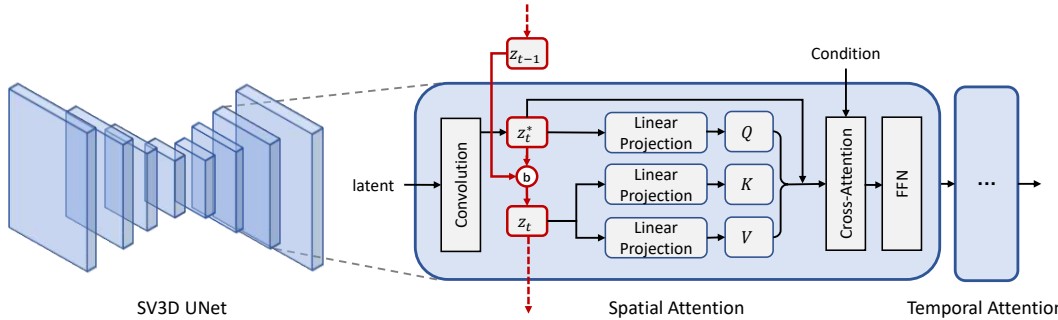

Figure 9: **Network details of Attention Injection.** ⓑ denotes the EMA blending operator mentioned in Sec. 4.1. $z_t^*$ is the multi-view latent at current timestamp $t$, and $z_t$ is the blended latent. Previous visual information is injected into the current latent by modifying the original spatial self-attention mechanism.

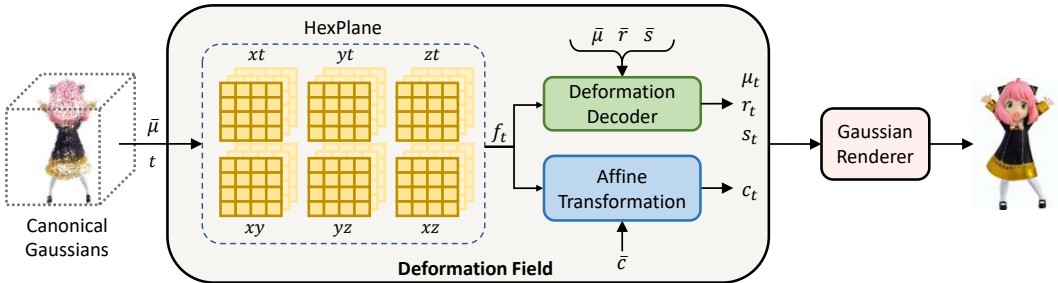

Figure 10: **Network architecture of our 4D representation.** $\bar{\mu}, \bar{r}, \bar{s}$ represents the canonical Gaussian properties: 3D location, rotation and scale from the coarse stage training in 4DGS (Wu et al., 2024a). The time-specific local feature $f_t$ is queried from the HexPlane (Cao & Johnson, 2023), where the subscribe $t$ means the time-specific property. Different from vanilla 4DGS, we employ additional color affine transformation to obtain the time-specific color $c_t$. The geometric deformations are predicted by a lightweight decoder. Finally, the time-specific Gaussians are rendered to produce an image (right).

## C.2 REPRODUCTION, DATA AND CODE

We reproduced our baselines, including Animate124 (Zhao et al., 2023), DreamGaussian4D (Ren et al., 2023), and Consistent4D (Jiang et al., 2024b), Efficient4D (Pan et al., 2024), using their official code. Additionally, we have included the input images and SVD-generated videos in the *supplementary materials*. Apart from the data provided by Animate124 and DreamGaussian4D, we have added three more examples: `android`, `chicken-basketball`, and `penguin`. Code is also available in the *supplementary materials*.

## C.3 USER STUDY DETAILS

We provide details of the user preference study with two screenshots. Figure 21 illustrates the guidelines: each participant is asked to evaluate images and videos rendered by four different methods across five metrics. Figure 22 shows the image and video samples presented to the participants. After comparing the images (Figure 22(a)) rendered by different models, participants select the method with the highest "reference image consistency" and "3D appearance". After watching the videos (Figure 22(b)) rendered by different models, participants select the method with the highest "motion realism" and "motion range". Finally, they choose the method with the best overall quality. We presented several cases to 47 participants and compiled the statistics. For statistical significance, we make the assumption of multinomial distribution, and report the 2-sigma error bar (95.6% CI). We use standard deviation for error bar calculation.

| Method | Independent | S-Res | S-Linear | T-EMA | S-EMA (Ours) |
|---|---|---|---|---|---|
| CLIP-I ↑ (Radford et al., 2021) | 0.9323 | 0.9136 | 0.9654 | **0.9962** | 0.9925 |
| Flow Intensity↑ (Teed & Deng, 2020) | - | - | **2.912** | 1.102 | 2.756 |

Table 4: **Quantitative ablation on attention injection.** We evaluate temporal consistency using CLIP-I score between the first and subsequent frames (↑ higher is better), and motion range using optical Flow Intensity (↑ larger indicates larger motion range when CLIP scores are comparable). '-' means no reasonable results predicted by the optical flow estimator on video with noisy background. Both 'T-EMA' and 'S-EMA' improve temporal consistency, but while 'T-EMA' results in nearly static output, 'S-EMA' maintains substantial motion range. Qualitative results are shown in Figure 5.

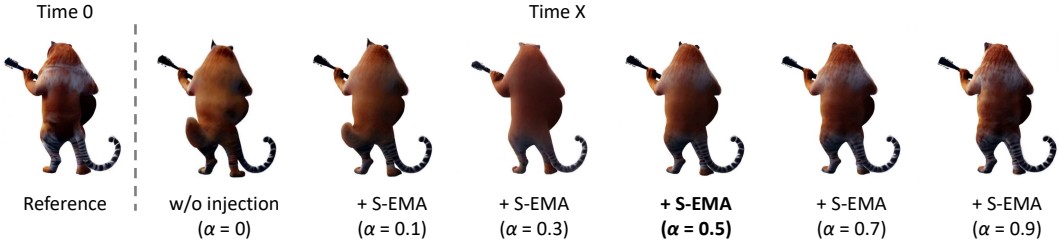

Time 0                                    Time X

Reference | w/o injection | + S-EMA | + S-EMA | **+ S-EMA** | + S-EMA | + S-EMA
          | ($\alpha$ = 0) | ($\alpha$ = 0.1) | ($\alpha$ = 0.3) | **($\alpha$ = 0.5)** | ($\alpha$ = 0.7) | ($\alpha$ = 0.9)

Figure 11: **Qualitative sensitivity analysis on EMA blending weight of attention injection**. As the blending weight increases, the temporal consistency is significantly improved (similar white textures and consistent leg geometry). However, the overly high ($> 0.5$) blending weight leads to a very small motion range. To balance the motion range and temporal consistency, we choose the EMA weight as $\alpha = 0.5$. Video demonstration can be found in our *supplementary materials*.

## D EXTENDED ABLATIONS

**Quantitative ablation on attention injection.** We conduct comprehensive experiments to evaluate different variants of attention injection. Please refer to Figure 5 in the main paper for qualitative results. Quantitative analysis is summarized in Table 4, demonstrating the impact of different attention injection variants on video temporal consistency and motion range. To quantify temporal consistency, we compute the CLIP-I score between the first frame and subsequent frames. Our results indicate that both 'T-EMA' and 'S-EMA' significantly improve temporal consistency (inter-frame similarity), achieving higher CLIP-I scores compared to other variants. For motion range assessment, we employ the flow intensity, calculated by average value of optical flow on adjacent frames. When CLIP scores are comparable, larger flow intensity indicates a larger range of motion. The optical flow is estimated with RAFT (Teed & Deng, 2020). '-' means no reasonable results predicted by the optical flow estimator on video with noisy background. Notably, 'T-EMA' yields a DINO score approaching 1, suggesting minimal object movement. Among all variants examined, our proposed 'S-EMA' uniquely achieves an optimal balance, maintaining high temporal consistency while preserving substantial motion range.

**Sensitivity analysis for attention injection weight.** Figure 11 analyzes different EMA blending weights $\alpha$ of attention injection in the spatial attention layers. It is obvious that the increasing blending weight benefits the temporal consistency in texture, *e.g.*, similar white texture in the back and consistent leg geometry. We also observe that overly high ($> 0.5$) blending weight significantly attenuates the object motion range. This trade-off can be better illustrated by the videos provided in the *supplementary materials*. Taking both motion range and temporal consistency into consideration, we choose $\alpha = 0.5$ as an appropriate blending weight without sacrificing the dynamics. Figure 12 demonstrates the impact of different blending weights. As the weight increases, CLIP-I score (image quality) improves while motion range becomes smaller, indicated by decreasing flow intensity. CD-FVD will not be better due to the diminishing motion.

**Number of Gaussians.** In Figure 13, we show the number of Gaussians before and after adding the multiscale renderer. Guo et al. (Guo et al., 2024b) observed that visual overfitting often leads to redundant Gaussian splats in dynamic scene reconstruction, which is hard to optimize and causes

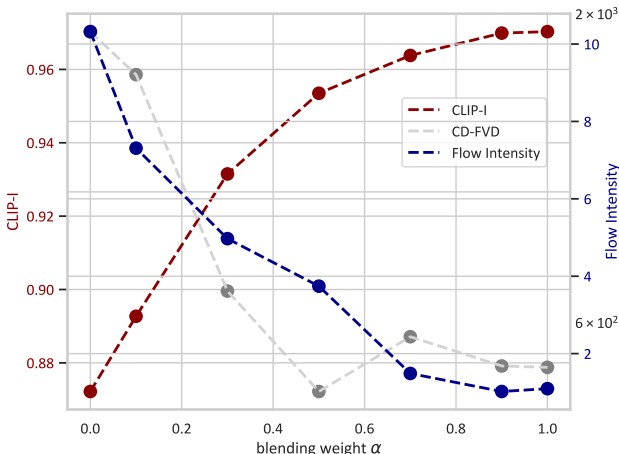

Figure 12: **Quantitative sensitivity analysis on EMA blending weight of attention injection.** Higher blending weights improve temporal consistency but excessive values restrict motion range indicated by lower flow intensity. We select $\alpha$=0.5, achieving well balance between motion range and temporal consistency, with the best CD-FVD score.

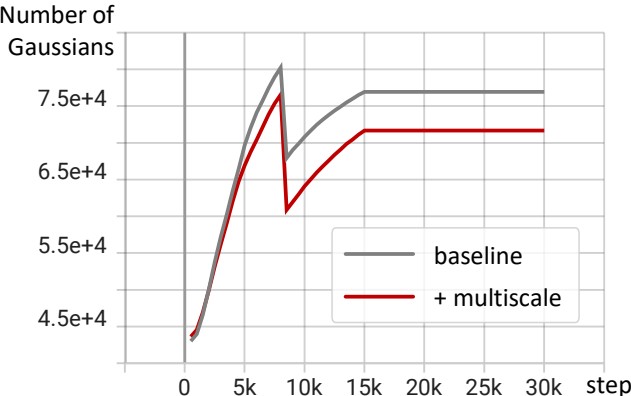

Figure 13: **Additional ablation on the multiscale renderer.** With the multiscale rendering augmentation in Stage II (darkred), the number of Gaussians declines significantly.

unsatisfying rendering results. With the multiscale renderer, we observe a significant decline of Gaussian points, in addition to the dropped training PSNR reported in Figure 7.

**Additional results for diffusion refinements.** In Figure 14, the effectiveness of our diffusion refinement is illustrated with zoomed-in details. It can be observed that the facial and hand details become finer and Gaussian noises are removed after the refinement stage.

## E    EXTENDED RESULTS

**Dynamics of our results.** For the best demonstration of our 4D model dynamics, please refer to the *supplementary materials* where you can find videos generated by our 4D model. Figure 15 has illustrated more examples beyond SVD-generated videos, and show the scalability and generalizability of our framework. The panel (a) uses video rendered from Objaverse (Deitke et al., 2023) dataset, a large-scale 3D dataset that also contains some animation models. Figure 15 (b) shows the 4D generation results from in-the-wild videos from the Consistent4D benchmark; In panel (c), we leverage the pose-conditioned character video generation model, AnimateAnyone (Hu, 2024), as our video model in our framework.

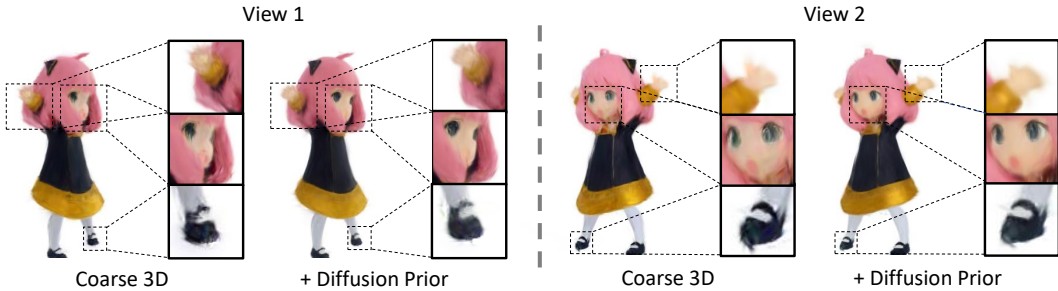

Figure 14: **Ablation on diffusion refinement.** The left and right panels depict two different view of renderings with the case `anya`. The results after adding the diffusion refinement show finer facial and hand details with less noisy Gaussians.

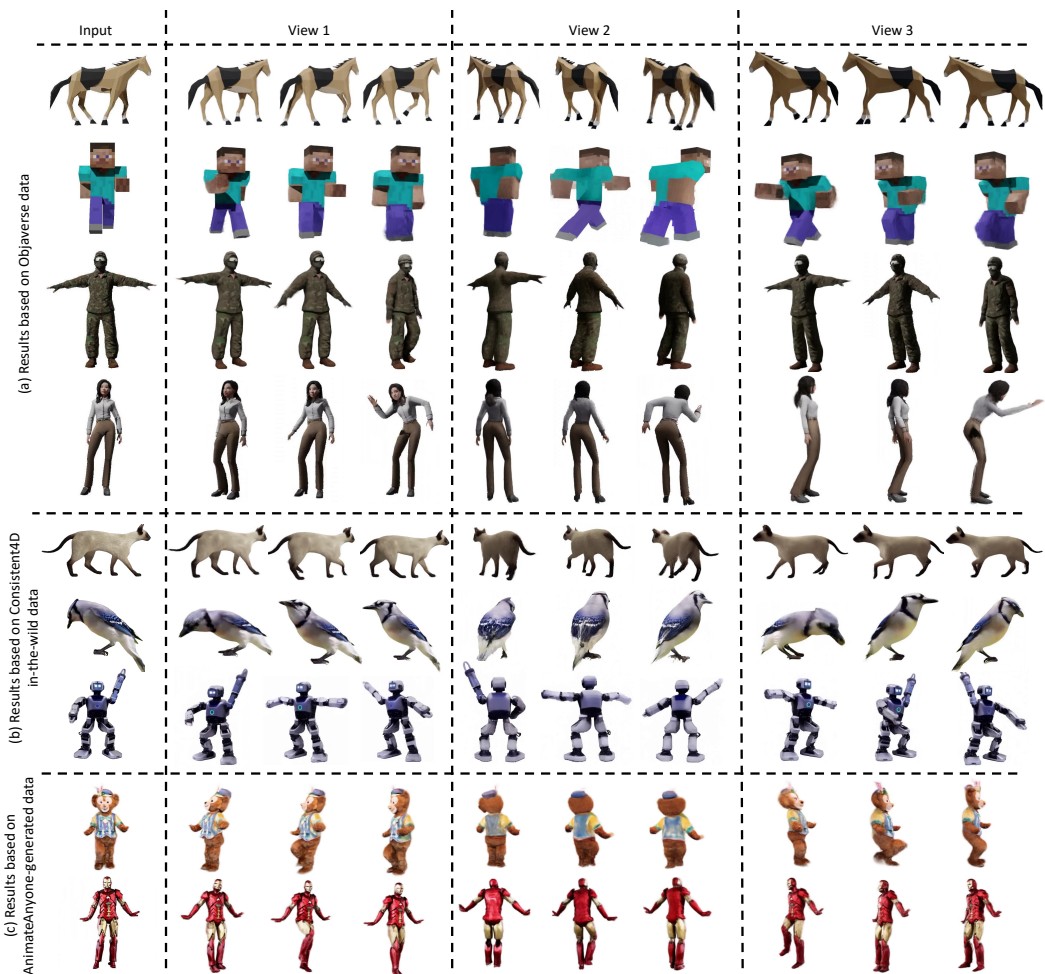

Figure 15: Extended results rendered by our EG4D based on semantic/large motion of synthetic/real-world objects. The input data includes **(a)** single-view rendering from Objaverse (Deitke et al., 2023) objects, **(b)** in-the-wild videos from Consistent4D (Jiang et al., 2024b), and **(c)** character motions generated by pose-conditioned video diffusion, AnimateAnyone (Hu, 2024).

**Qualitative comparison with Efficient4D.** We compare our results with another baseline Efficient4D (Pan et al., 2024), which uses 4DGS (Yang et al., 2024a) as reconstruction backbone. Consistent with quantitative results in main paper Figure 2, the 4D results generated by our method have higher view consistency and temporal consistency.

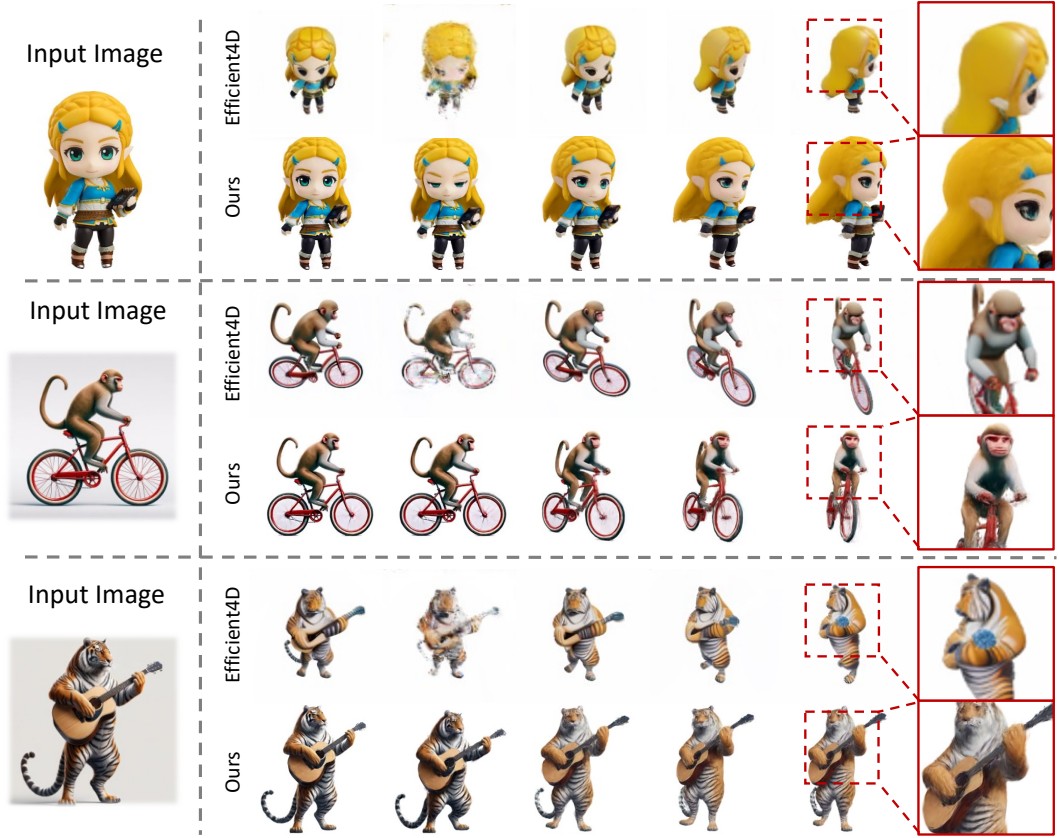

Figure 16: Qualitative comparison with another baseline, Efficient4D (Pan et al., 2024).

| Method | CLIP-I↑ | PSNR↑ | SSIM↑ | LPIPS↓ | FVD↓ | CD-FVD↓ |
|---|---|---|---|---|---|---|
| SV4D (Xie et al., 2024) | 0.9459 | 22.57 | 0.852 | 0.196 | **138.81** | **311.06** |
| EG4D (Ours) | **0.9535** | **23.28** | **0.904** | **0.173** | 142.34 | 459.10 |

Table 5: Quantitative comparison with a training-based method, SV4D (Xie et al., 2024).

**Comparison with training-based methods.** Recent works (Liang et al., 2024; Jiang et al., 2024a; Xie et al., 2024) have advanced multi-view video diffusion through training on large-scale 4D datasets, demonstrating significant improvements in 4D generation quality. Notably, Animate3D (Jiang et al., 2024a) extends AnimateDiff (Guo et al., 2024a) to generate spatiotemporally consistent multi-view videos of static 3D objects. We compare our method with SV4D (Xie et al., 2024), as shown in Figure 17 and Table 5. While SV4D achieves better temporal consistency, our approach exhibits superior image fidelity and view-consistency. This is evident in examples like `luigi` and `zelda`, where SV4D produces overly bright faces lacking detail and shading. This suggests that while SV4D performs well on its training set, it may have limited generalization capability on out-of-distribution (O.O.D.) samples.

**More discussion about training-based methods.** Our framework offers two key advantages over training-based methods like Diffusion4D and SV4D: First, our approach is *training-free*, leveraging off-the-shelf video and multi-view diffusion models without modifications. This allows rapid adoption of advances in either model type to generate 4D content efficiently, eliminating the need for expensive training on large-scale 4D datasets. Second, our method maintains *dataset independence* and directly benefits from improvements in video diffusion. Regarding motion of 4D object, advanced video diffusion models like CogVideoX (Yang et al., 2024b) would enable more dynamic and diverse animations. For 3D content, multi-view diffusion for 3D scene, *e.g.* ViewCrafter (Yu et al., 2024), would provide possibility for 4D scene generation.

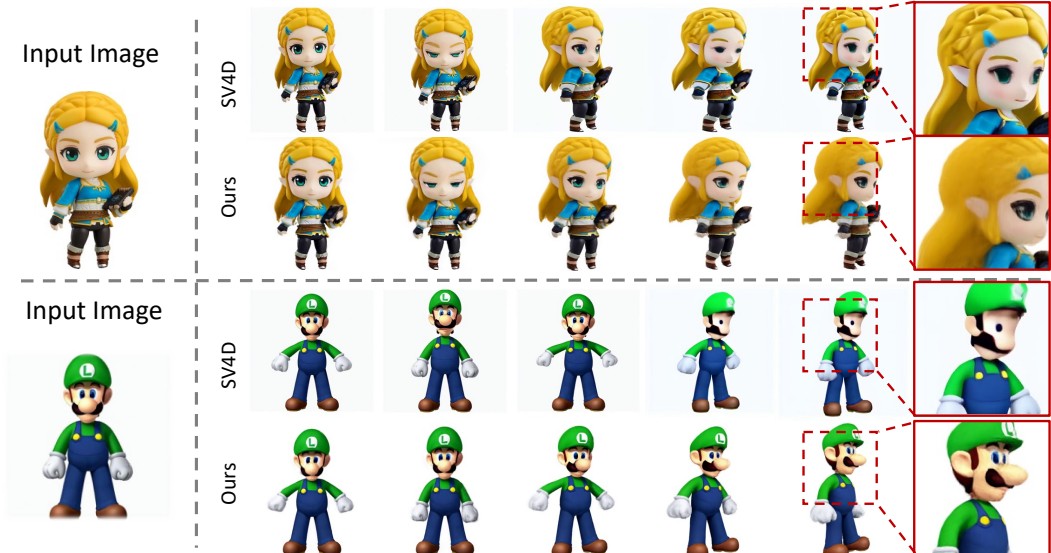

Figure 17: **Comparison with Training-based method, SV4D (Xie et al., 2024).** Benefited from the dataset-independency, we achieve higher view-consistency compared to SV4D in O.O.D data.

**4D Generation with complex image prompt.** Apart from the images/videos from Animate124 and Consistent4D benchmark, we experiment our methods in more complex image prompt from DreamCraft3D (Sun et al., 2023). For images with extreme lighting (Figure 18 first row) and complex image prompts (Figure 18 row 2-4), we find that our method can produce results with high view-consistency and image fidelity. However, we find it hard to achieve stable results for some complex images. The main difficulties come from image-based video generation models: SVD or even the state-of-the-art video generation, CogVideoX-I2V (Yang et al., 2024b), can not always produce videos with well-preserved identity and smooth motion. Generated videos, corresponded demo videos, and failure generated videos are all included in the supplementary material.

**Efficiency.** Our framework takes approximately 1 hour and 15 minutes on average for each 4D object generation in a single NVIDIA RTX 3090. Specifically, Stage I requires about 20 minutes for video and multi-view generation; Stage II, involving 4D Gaussian Splatting optimization, takes around 25 minutes; and the refinement process takes about 30 minutes. In previous works, Consistent4D (Jiang et al., 2024b) and Animate124 (Zhao et al., 2023) take about 2.5 and 9 hours, respectively, for 4D generation. Notably, DreamGaussian4D (Ren et al., 2023) achieves extremely short optimization time of 7 minutes. Diffusion4D (Liang et al., 2024) and SV4D (Xie et al., 2024) train a diffusion network to generate the multi-view multi-frame image matrix. We have comparable inference time since both methods share similar pipeline of multi-view video generation and 4D representation optimization. However, they need to take huge computational expense for training. In contrast to optimization-based approaches, L4GM (Ren et al., 2024) uses feed-forward network to direct predict the Gaussian sequences within several minutes. Our optimization time falls between these, but our framework offers superior view consistency, 3D appearance, and motion quality. Since Stage I appears to be one of the efficiency bottlenecks, future work should focus on incorporating efficient sampling for video diffusion models to boost speed.

**Multi-view results of our results.** Figure 23 shows the multi-view results of our 4D model, which is a supplement of Figure 4. Due to the page limit of the main paper, we only show two views of the 4D model there, which is not enough to illustrate the 3D appearance of our model. To this end, we render our model in more views: 0°, 90°, 135°, 180°, 225°, and 270°. The rendered multi-view images show that our method can produce images with high 3D consistency and satisfactory quality.

**More visual comparisons.** Figure 24 provides additional visual comparisons with our baselines, continuing from Figure 3 in the main paper. We use three additional cases: `luigi`, `anya`, and `chicken-basketball`. The first two columns show animation results from the same view,

Input Image                                        4D Rendering

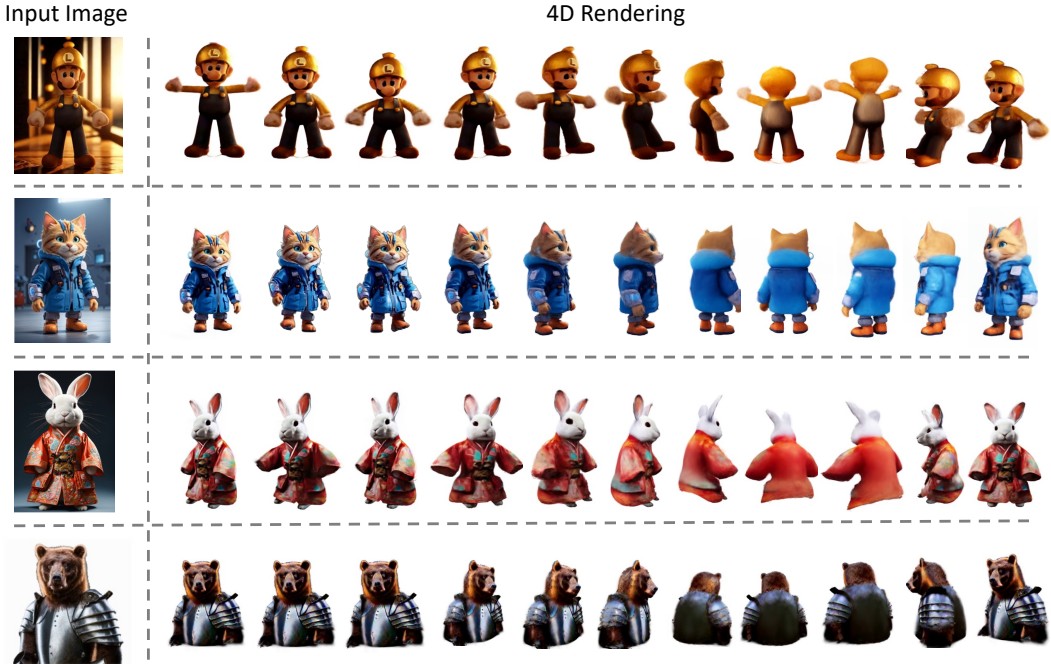

Figure 18: **4D generation results with complex image prompts.** The first row is `luigi` relighted by IC-Light (Zhang et al., 2024), and 2-4 rows are image prompts from DreamCraft3D (Sun et al., 2023). Since SVD (Blattmann et al., 2023a) fails to generate valid video for `bear-armor` case, we have opted to utilize CogVideoX-I2V (Yang et al., 2024b) as an alternative.

Text Prompt:
"*ninja, white background, standing, toy, cartoon, 3D model, high quality*"

Generated Image

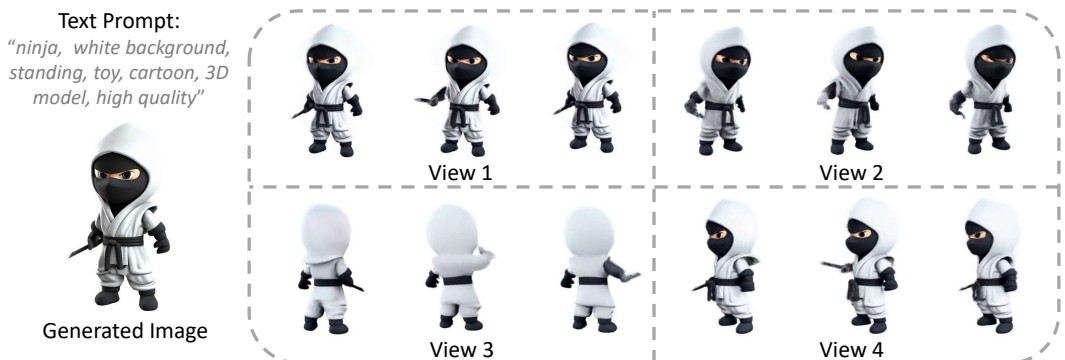

Figure 19: **Text-to-4D results.** We feed a text prompt (left top) into SDXL Podell et al. (2024) to generate a ninja image (left bottom). This image can be transformed into 4D objects with our framework, presenting indirect text-to-4D application. The right panel shows multi-view renderings of the 4D model.

while column 3 to 5 display three different views. The last column presents a zoomed-in image of the final rendered view. Multi-view videos for visual comparison can be found in the *supplementary*.

**More applications.** Benefiting from our explicit generation, we can easily adapt EG4D to both text-to-4D and video-to-4D tasks. Figure 19 shows the generation results of the text-to-4D. We first feed an example text prompt into SDXL (Podell et al., 2024) to get the high-resolution image. Then this image is transformed into a 4D model with our framework. Figure 20 shows the results of the video-to-4D. We just skip the dynamic generation step and start with our view synthesis pipeline.

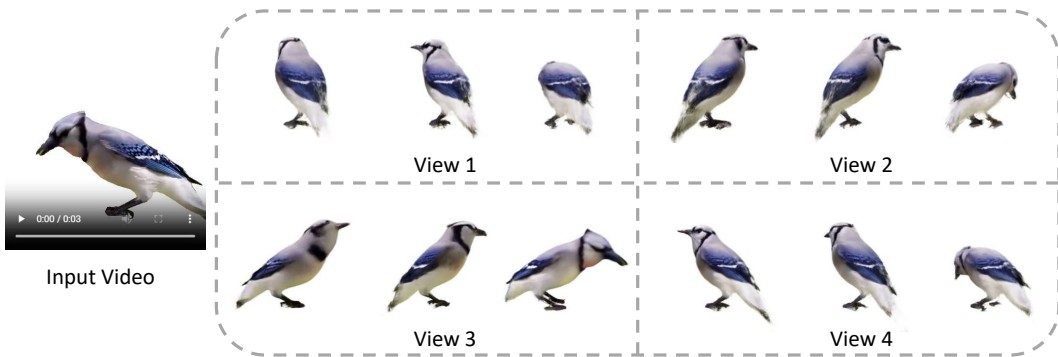

Figure 20: **Video-to-4D results.** Our framework can be seamlessly extended to video-to-4D generation. The right panel shows the renderings of our 4D model from four viewpoints. This `bird` video is taken from Consistent4D (Jiang et al., 2024b).

First of all, thank you all for participating!

Our task is to generate a 4D model from a given image, and then render it at arbitrary view/time.

Please compare the generation results produced by different methods and answer the following questions.

First, please compare the images produced by four methods and select one method that you think provide the best results.

◆ Which method's results have better consistency with the given image?
    • *Focus on consistency instead of quality*
◆ Which method's results have the best 3D appearance?
    • *Focus on esthetics and view-consistency*

Then, please compare the videos produced by those methods.

◆ Which method produces the most natural motion?
◆ Which method produces the largest range of motion?

Finally,

◆ Please select the method that shows the best overall quality!

Figure 21: **Screenshot of our user study guidelines.** Each participant is asked to evaluate the images and videos rendered by 4 different methods with 5 metrics, *i.e.*, reference view consistency, 3D appearance, motion realism, motion range, and overall quality.

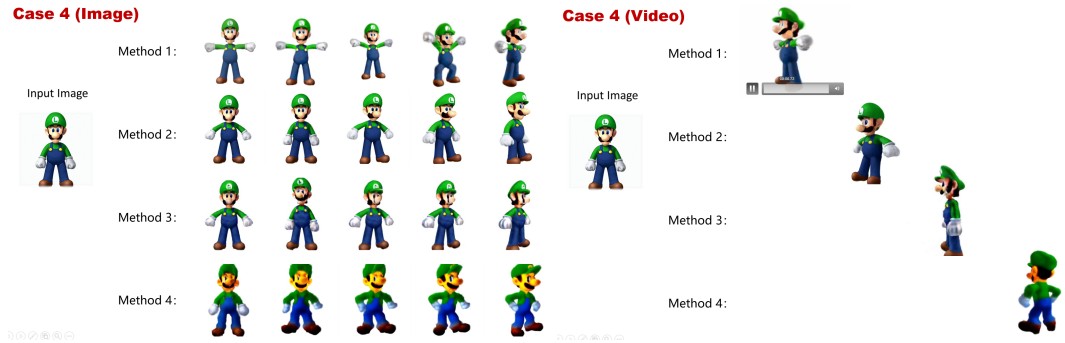

      (a) Screenshot of the image evaluation.             (b) Screenshot of the video evaluation.

Figure 22: **Screenshot of our user study content.** Each participant is provided with several images and videos rendered by different methods.

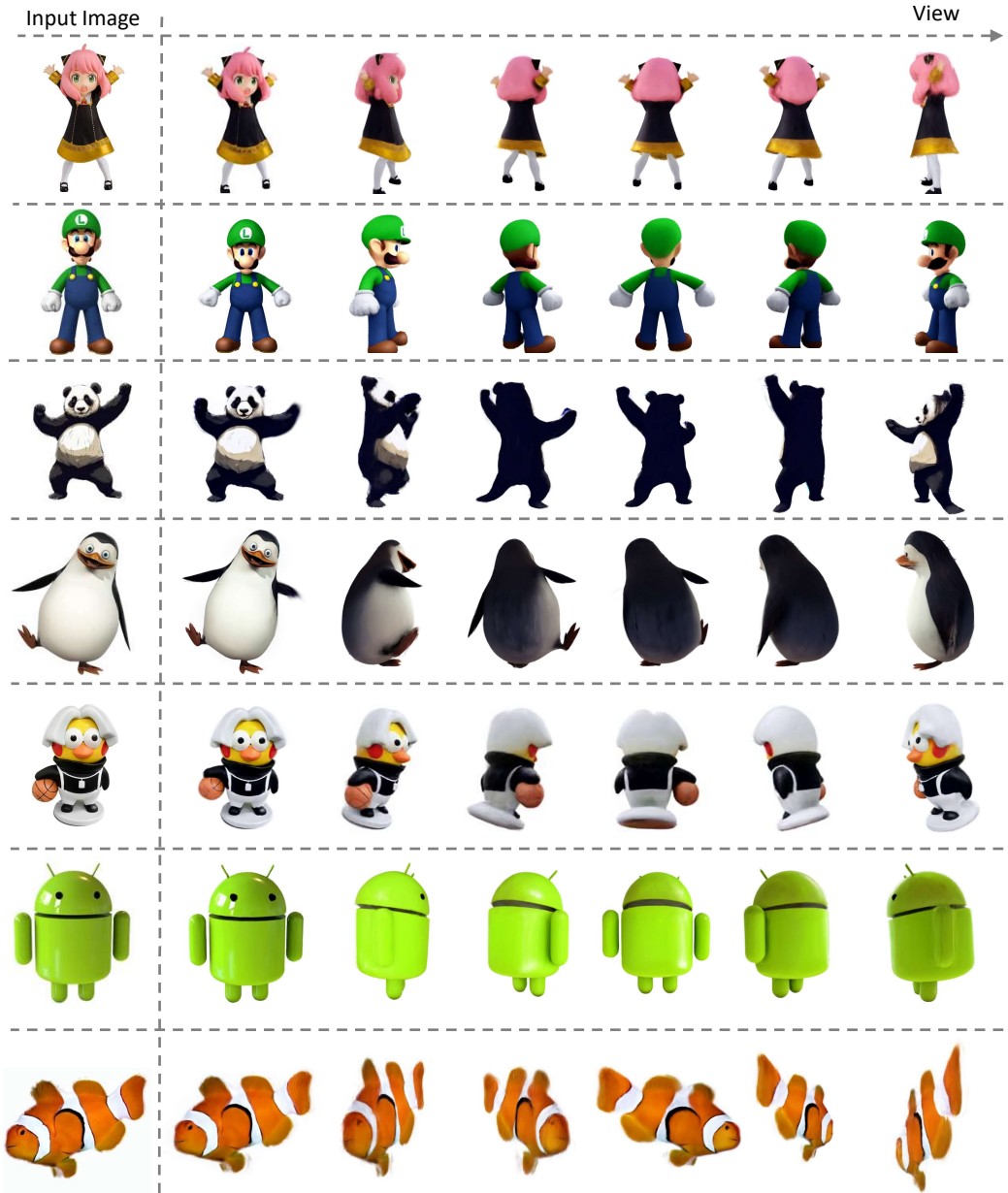

Figure 23: **Multi-view results of our models.** This figure is a supplement of Figure 4 in the main paper. The 6 columns show the images rendered by our model in different views: 0°, 90°, 135°, 180°, 225°, and 270°. Multi-view renderings demonstrate the geometry/texture consistency and promising quality of our 4D representation.

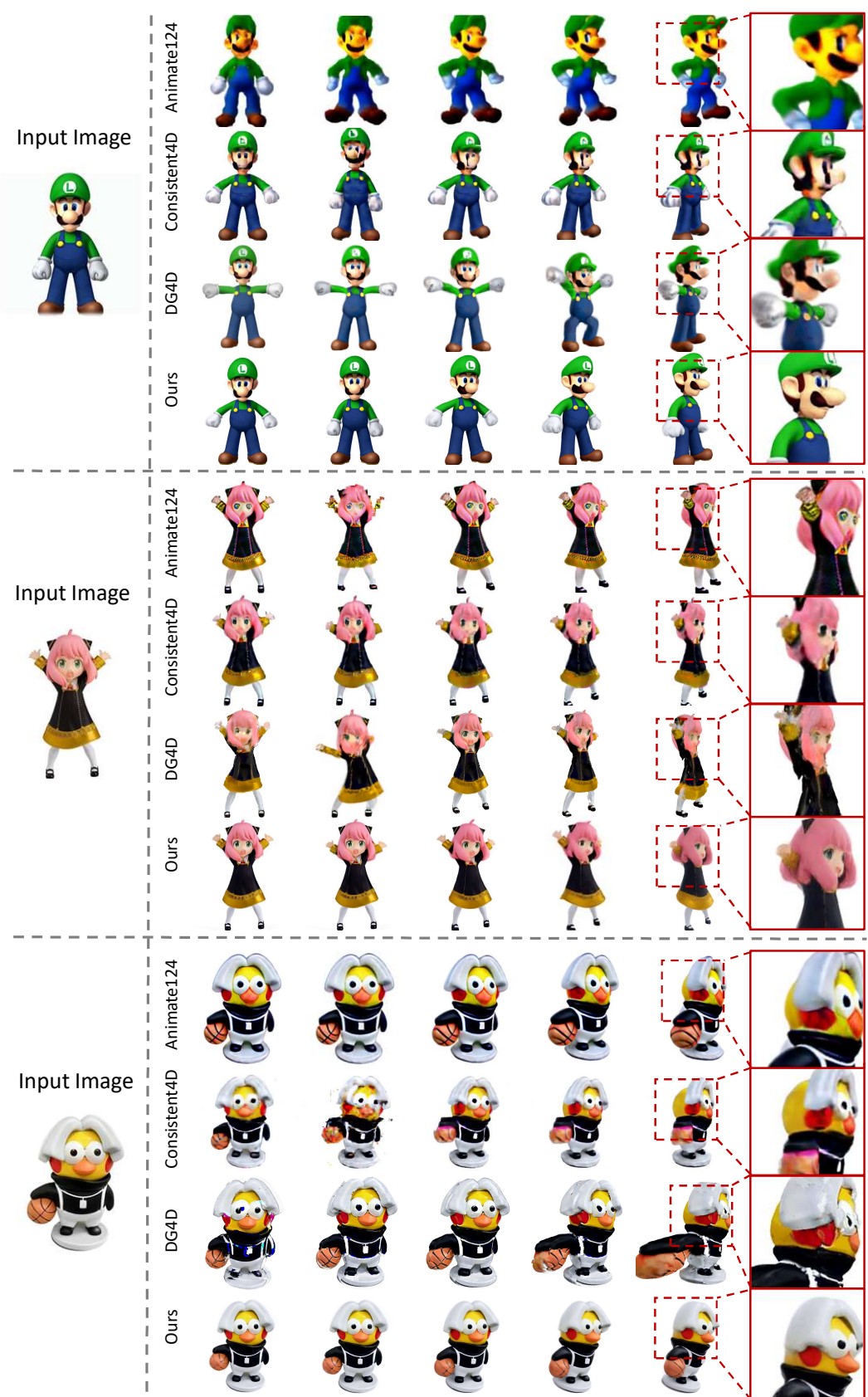

Figure 24: More comparison examples with the SOTA results in three cases `luigi`, `anya` and `chicken-basketball` (better zoom in).

