# OpenReview forum: "EG4D: Explicit Generation of 4D Object without Score Distillation"
_ICLR.cc/2025/Conference — ICLR 2025 Poster_

### Official Review · Reviewer_5QU4 · 2024-10-30

**Soundness:** 3
**Presentation:** 3
**Contribution:** 3
**Rating:** 6
**Confidence:** 4

**Summary:**

The paper proposes an explicit 4D object generalization method without using SDS loss. The method leverages SVD and SV3D to optimize the 4D representation from the generated videos. Experimental results demonstrate superior performance over previous baselines.

**Strengths:**

1.	The proposed pipeline is clearly motivated and reasonable.
2.	The writing is clear and easy to follow, with a well-structured introduction and related work section
3.	Extensive experiments shows clear performance improvements over previous methods (Table 1, Table 2).
4.	The authors provide a comprehensive ablation study, which helps justify the effectiveness of attention injection, color transformation, multi-scale renderer, refinement strategies.
5.	The supplementary materials are sufficient.

**Weaknesses:**

1.	As shown in the video, the results are flickering and blurry. It seems that the quality is not as good as Diffusion4D [1] (see the examples of the project page of Diffusion4D).  It would be valuable to add a discussion and analysis about the proposed method with Diffusion4D (e.g., generation time, temporal consistency, image fidelity, 3D consistency).

2.	The method is built on SVD and SV3D for 4D generation. It would strengthen the paper if the authors could discuss the different designs between SV4D [2] and proposed method. The authors are suggested to provide both qualitative and quantitative comparisons. In specific, the authors can quantitative comparison table between their method and SV4D on metrics like overall quality (view alignment, 3D consistency, multi-view consistency,  temporal consistency), animation flexibility (e.g. motion range, motion fidelity), and computational efficiency (e.g. FPS, generation time).

3.	I noticed the examples are some simple images. I wonder if the proposed can handle more complex images like examples in https://mrtornado24.github.io/DreamCraft3D/ and other cases (with multiple objects, complex backgrounds, or challenging lighting conditions).

[1] Liang, Hanwen, Yuyang Yin, Dejia Xu, Hanxue Liang, Zhangyang Wang, Konstantinos N. Plataniotis, Yao Zhao, and Yunchao Wei. "Diffusion4D: Fast Spatial-temporal Consistent 4D Generation via Video Diffusion Models." arXiv preprint arXiv:2405.16645 (2024).
[2] Xie, Yiming, Chun-Han Yao, Vikram Voleti, Huaizu Jiang, and Varun Jampani. "Sv4d: Dynamic 3d content generation with multi-frame and multi-view consistency." arXiv preprint arXiv:2407.17470 (2024).

**Questions:**

Another way is to first generate 3D models and then animate the mesh [4]. I wonder if this line of methods are more effective and flexible.  Could you discuss the potential advantages and limitations of their approach compared to the 3D-then-animate pipeline. Specifically, the authors are  suggested they address aspects like computational efficiency, animation quality, and flexibility in handling different types of objects or motions.

[4] Jiang, Yanqin, Chaohui Yu, Chenjie Cao, Fan Wang, Weiming Hu, and Jin Gao. "Animate3d: Animating any 3d model with multi-view video diffusion." arXiv preprint arXiv:2407.11398 (2024).

---

> ### Author Response · Authors · 2024-11-22
> **Response to Reviewer 5QU4 (1/2)**
>
> We appreciate the reviewer for encouraging feedback and discussion about several concurrent training-based methods.
>
> **Q1: Discussion about training-based method (Diffusion4D, SV4D)**
>
> - Difference in method
>
> Diffusion4D collects a large-scale 4D dataset from Objaverse. Based on the 3D video diffusion, they train a 4D-aware video diffusion model with motion magnitude guidance. Specifically, they assert temporal layers on the multi-view diffusion to model the spatial-temporal consistency in orbital views of 4D objects.
>
> Similar to Diffusion4D, SV4D also collects a large-scale 4D dataset from Objaverse, named ObjverseDy. They finetune SV3D as the multi-view model, inserting temporal (frame) layers initialized with SVD. Due to limitation of memory, they first sample a sparse set of anchor frames, then sample the middle frames conditioned on the anchor frames.
>
> Our methods as a training-free method has no need to collect and train on 4D dataset, and thus will not limited to the training object categories. Ours just samples multi-view frames with attention injection in SV3D spatial layer (S-EMA) to generate multi-view videos, which is more flexible and generalizable.
>
> For 4D representation, SV4D uses dynamic NeRF, while Diffusion4D and our method use 4DGS.
>
> - Empirical comparison
>
> However, Diffusion4D does not release the inference code. Thus, we take the opensourced SV4D as a representative of these methods, since the core idea and training dataset are similar. We compare our method with SV4D in **Appendix Figure 17 and Table 5**, explained as follows:
>
> 1. Multi-view alignment & image fidelity. Our method achieves superior performance in CLIP-I score and novel view synthesis metrics. Clear quality differences are visible in Animate124 samples (`luigi` and `zelda`), where SV4D produces overly bright faces lacking detail and shading, and exhibits lower view-consistency. These results suggest SV4D may have limited generalization capability on non-Objaverse data or out-of-distribution (O.O.D.) samples.
> 2. Motion range & motion fidelity. SV4D shows marginally better performance in video quality metrics and temporal consistency. It achieves slightly larger motion range with the same video input, likely due to the attention injection for smoothing the dynamics.
> 3. Comparable generation time: SV4D and our method both need multi-view video generation and 4D representation optimization.
>
> **Table R3.1**: Comparison with SV4D in Animate124 benchmark.
>
> |  | CLIP-I | PSNR | SSIM | LPIPS | FVD | CD-FVD |
> | --- | --- | --- | --- | --- | --- | --- |
> | SV4D | 0.9459 | 22.57 | 0.852 | 0.196 | **138.81** | **311.06** |
> | Ours | **0.9535**  | **23.28** | **0.904** | **0.173** | 142.34 | 459.10 |
> - More discussions about training-based method
>
> There are two term of advantages of our framework over training-based methods (Diffusion4D, SV4D):
>
> (1) Training-free approach: Our method leverages off-the-shelf video and multi-view diffusion models without modifications. This design allows immediate adoption of advances in either diffusion model, enabling 4D generation within acceptable time consumption, at least for preview purposes. We avoid the need for expensive training on large-scale 4D object datasets for multi-view video diffusion.
>
> (2) Dataset independence: Our method's flexibility enables broader applicability and ready adaptation to advances in diffusion models:
>
> Motion independence: Our framework can directly benefit from improvements in video generators (e.g., CogVideoX) to produce more dynamic and diverse motions.
>
> (3) 3D content independence: Advances in multi-view diffusion models (e.g., ViewCrafter) can be integrated to potentially extend our approach to 4D scene generation.
>
> **Q2: Can you experiment on complex images? Can EG4D tackle the background/multiple objects and extreme lighting condition?**
>
> In the current version, we have provided the images in DreamCraft3D in **Appendix** **Figure 18**.
>
> - Background: We remove the background as preprocessing. Figure 18 (up) shows the generation results with complex background.
> - Extreme lighting condition: We take the `luigi` relighted with IC-Light [1] as the illustrative case. Figure 18 (down) shows that our method has the capability for the image prompt with extreme lighting condition. Two generated videos are included in the supplementary material.
> - Multiple objects: Our multi-view generation ability comes from SV3D, so it would be hard to tackle multiple objects.

---

> ### Author Response · Authors · 2024-11-22
> **Response to Reviewer 5QU4 (2/2)**
>
> **Q3: Advantages and limitations of “animate-then-lift” compared to the “3D-then-animate” pipeline. Can you discussion about Animate3D?**
>
> We acknowledge the reviewer's reference to the concurrent work Animate3D, which extends the video generation model AnimateDiff to multi-view images. While Diffusion4D/SV4D processes monocular video input, Animate3D takes a different approach by using multi-view renderings as input to generate temporally-coherent multi-view videos.
>
> Comparison between “animate-then-lift” and "3D-then-animate" pipeline:
>
> - Advantages:
>
> (1) "3D-then-animate" first uses **single-view image** for 3D reconstruction, which would be challenging. Compared to “3D-then-animate”, “animate-then-lift” pipeline first generates single-view video, then joint learning from the multiple video frames that contains varying motions, which could provide comprehensive clues and constraints for 3D modeling.
>
> (2) For "3D-then-animate" pipeline, animation of static posed 3D models reconstructed in first stage is difficult, especially for complex poses or self-occluding cases. For example, when given an image of a person with crossed arms against their chest, distinguishing between the arms and torso becomes problematic. Our approach (”animate-then-lift”) benefits from observing various poses during animation, enabling more accurate reconstruction of the canonical 3D shape.
>
> (3) It would be easier for “animate-then-lift” pipeline to generate more diverse and complex motions, since single-view video generation is easier than multi-view video generation.
>
> - Limitations:
>
> “Animate-then-lift” pipeline involves joint optimization of 4D representation, which would be more computational intensive. By separating 3D modeling from animation, "3D-then-animate" pipeline can focus specifically on maintaining multi-view consistency during video generation, which would be easier to optimize and better consistency.
>
> [1] https://github.com/lllyasviel/IC-Light

---

> ### Comment · Reviewer_5QU4 · 2024-11-26
>
> Thanks for your detailed feedback. For Q2 (complex images, I observe the provided video
>  demonstrates only a slight viewpoint range. To better showcase the proposed method's capability, it is recommended to include results with more extensive viewpoint variations. A commonly-used approach in 4D generation is to present turn-around videos, as seen in [example](https://jiawei-ren.github.io/projects/dreamgaussian4d/videos/examples.mp4) in [DreamGaussian4D]. Additionally, it would be beneficial to provide more than one example of complex images accompanied by turn-around videos to demonstrate the method's robustness and effectiveness. Suggested examples include   ["bear_armor"](https://mrtornado24.github.io/DreamCraft3D/assets_dreamcraft3d/videos/bear_armor.mp4), ["Messi"](https://mrtornado24.github.io/DreamCraft3D/assets_dreamcraft3d/videos/messi.mp4), and ["boar"](https://mrtornado24.github.io/DreamCraft3D/assets_dreamcraft3d/videos/boar.mp4). This will strengthen the impact of the results and provide a more comprehensive evaluation.

---

> ### Author Response · Authors · 2024-11-26
> **Response to Reviewer 5QU4 (2nd Round)**
>
> Dear Reviewer 5QU4,
>
> Thank you for suggestions on result demonstration for complex image prompts. In the last version, we have only included the **SVD-generated single-view video** in the supplementary.
>
> In the current version, we have revised **Figure 18**, including **turn-around videos** of `luigi-relighted`, `cat` cases (in the last version of paper), and two new cases `rabbit` and `bear-armor`  cases from DreamCraft3D.  We find that our result can generate results with good view-consistency and image fidelity. However, we find it hard to achieve stable results for some complex images, like `messi` and `boar`. The main difficulties come from image-based video generation models: SVD or even the state-of-the-art video generation, CogVideoX-I2V, can not always produce videos with well-preserved identity and smooth motion.
>
> Additionally, all turn-around videos are included in the sub-directory `videos/complex/demo-complex.mp4`  of the **supplementary**. The failure single-view videos are also included in `videos/complex/failure-case-video-gen.mp4`. Revised part in manuscript (**Line 1160-1164 & Line 1210-1213**) is marked with red.
>
> Thank you once again for your encouraging comments and support in improving our work.
>
> Best Regards,
>
> Paper 5395 authors

---

> > ### Comment · Reviewer_5QU4 · 2024-11-27
> >
> > Thank you for the detailed response and the significant efforts to address my  suggestions. I also appreciate your candid discussion of the challenges faced with more intricate image prompts like messi and boar.

---

### Official Review · Reviewer_pq5e · 2024-11-03

**Soundness:** 2
**Presentation:** 3
**Contribution:** 2
**Rating:** 6
**Confidence:** 5

**Summary:**

The paper proposes a 4D generation framework that avoids the need of score distillation. Instead, multi-view videos are generated from one input image and a 4D representation is optimized directly.

**Strengths:**

- A multiscale augmentation is proposed for rendering more details.
- A training-free attention injection strategy is introduced to ensure consistency.
- A Diffusion Refinement stage is included to refine the details.

**Weaknesses:**

- In Fig. 1, it seems that the multiscale renderer is not used for the "Diffusion Refinement" stage. What is the motivation for this change? Is it necessary?
- What is the difference of the proposed Diffusion Refinement stage and that in DreamGaussian4D?

**Questions:**

- For the main comparison results. Did the authors use the same video to compare the proposed method and Dreamgaussian4D? It seems that the generated object motions are very different. If not, is it possible to additionally compare a variant of DreamGausisan4D? In other words, can we use the foreground video rendered by the proposed method and reconstruct to 4D with the framework of DreamGausisan4D?

- For the Mario case, as shown in the supplementary video, contains flickering artifacts on the face regions. Why is this happening? Is it because the refinement stage suffers from the Janus issue?

---

> ### Author Response · Authors · 2024-11-22
> **Response to Reviewer pq5e**
>
> We appreciate your valuable feedback, especially for pointing out detailed comparison with the pioneering work in 4D generation, DreamGaussian4D. Regarding your concerns, we address these as follows:
>
> **Q1: Can we apply multi-scale renderer on the Stage III?**
>
> As shown in Figure 1, the multi-scale renderer is not used in the "Diffusion Refinement" stage (Stage III) due to both design considerations and empirical findings. The goal of Stage III is to refine semantic details using Image-to-Image Diffusion. We observed that SDXL-Turbo performs best at high resolutions, while refinement at lower resolutions produces negligible differences compared to the original image, making the benefits of a multi-scale approach limited in this stage.
>
> Additionally, we conduct experiments to evaluate the use of a multi-scale renderer in Stage III and find no significant differences in performance or visual quality compared to using a single-resolution renderer, hence in current version we don't adopt the multi-scale renderer.
>
> **Table R2.1**: whether uses the multi-scale renderer in stage III.
>
> | Method | PSNR | SSIM | LPIPS | CLIP-I |
> | --- | --- | --- | --- | --- |
> | w/o Multi-scale renderer | **23.28** | **0.904** | 0.173 | **0.9535** |
> | w Multi-scale renderer | 23.10 | 0.899 | **0.170** | 0.9479 |
>
> **Q2: Difference with DreamGaussian4D Stage III.**
>
> - Difference in motivation: Our refinement aims to eliminate semantic defects in multi-view video generation, whereas DreamGaussian4D focuses on addressing temporal consistency.
> - Difference in method: DG4D employs SVD (a video diffusion model) to add-noise-denoise for random-view rendered videos.  But SVD has limitations in frame-level video quality and resolution. In contrast, we utilize SDXL-Turbo (an image diffusion model) to refine multi-view images for direct supervision of 4D Gaussian training, which has better image-level refine ability. Additionally, DG4D uses the refined videos to supervise texture image optimization with fixed mesh sequences. Compared to texture-optimization-only method, our method adopts joint refinement of geometry and appearance in 4DGS that yields superior visual results.
> - Experiment: We further explore the video temporal refinement in our framework. In the revised paper, we have included this video temporal refinement method as our ablation in **Figure 8**. The result in Figure 8 (e) shows that the video temporal refinement can hardly correct the semantic defect as ours. The face region is still blurred and lack details, and color shifting appears compared with the original image.
>
> **Q3: Comparison with variant of DreamGaussian4D.**
>
> Thanks for pointing out this for more fair comparison. We do not use the same video in the original version, and we just keep the input image the same as an image-to-4D method. In the revised paper, we have included  results of DG4D’s variant (video-based) in main paper **Figure 3**, and stated that in **Line 409**. The video we used for DG4D is included in the supplementary.
>
> **Q4: Artifacts in face regions of Mario? Is it caused by Janus problem?**
>
> The artifacts seen in the Mario example result from persistent temporal inconsistencies, not the Janus problem. Similar yet more severe artifacts are also evident in Consistent4D and Animate124. Our method has notably mitigated these issues when compared to these prior approaches. We have included an additional comparison video `comparison-diagonal-matrix-eg4d.mp4` in the **supplementary material** that demonstrates how our method enhances temporal consistency compared to the pseudo-video constructed from diagonal images of original SV3D+SVD image matrix.
>
> Regarding the Janus problem:
>
> This issue [1] typically arises from the orientation-agnostic nature of general image diffusion models and strong text guidance in SDS.  In Stage III, we use only one-step denoising and strong constraints of the original image guidance. Therefore, the observed artifacts are not brought by Janus problem.
>
> [1] Shi et al., MVDream: Multiview Diffusion for 3D Generation, ICLR 2024.

---

> ### Author Response · Authors · 2024-11-28
> **Looking forward to your feedback**
>
> Dear Reviewer pq5e,
>
> We sincerely appreciate your valuable feedback and suggestions. In our rebuttal, we have provided ablation on usage of multi-scale renderer in Stage III, precise comparison with (video-based) DreamGaussian4D, difference in refinement stage of DreamGaussian4D, and our improvement on temporal consistency.
>
> We kindly inquire whether these results and clarifications have adequately addressed your concerns. Please feel free to let us know if you have any further questions.
>
> Thank you very much!
>
> Best regards,
>
> Authors of Paper 5395

---

> > ### Comment · Reviewer_pq5e · 2024-12-01
> >
> > Thank you for the response! My concerns are resolved and I am happy to raise the rating.

---

> ### Author Response · Authors · 2024-12-02
> **Official Comment by Authors**
>
> Thanks for your suggestions and encouraging comments for improving our paper!

---

### Official Review · Reviewer_gHhE · 2024-11-03

**Soundness:** 2
**Presentation:** 3
**Contribution:** 3
**Rating:** 6
**Confidence:** 4

**Summary:**

The paper introduces EG4D, a multi-stage framework designed for generating 4D objects from a single image input without relying on score distillation sampling (SDS). EG4D employs three key stages: view generation using video diffusion models, coarse 4D reconstruction using Gaussian Splatting, and refinement with diffusion priors. The framework addresses common issues like temporal inconsistencies, over-saturation, and the Janus problem, achieving superior performance over existing 4D generation models. The authors demonstrate the effectiveness of EG4D through both qualitative and quantitative analyses, showing significant improvements in 4D object realism and consistency.

**Strengths:**

1. EG4D introduces a unique multi-stage approach that successfully avoids issues associated with score distillation, such as over-saturation and Janus artifacts.
2. The attention injection mechanism helps ensure temporal consistency.
3. Quantitative and qualitative results demonstrate that EG4D produces high-quality 4D content, achieving superior alignment with the reference view and more realistic motion realism compared to baselines like DreamGaussian4D and Animate124.
4. The evaluation is comprehensive. The evaluation metrics, including CLIP-I, PSNR, SSIM, FVD, and CD-FVD, along with a user study, offer robust evidence of the framework's advantages.

**Weaknesses:**

1. Lack of baseline methods. Recent approaches such as L4GM and efficient4D ( which also leverage 4D Gaussian Splatting) should definitely be included for comparison.
2. Lack of Comparative Analysis. The attention injection technique is novel, but without comparisons to other methods, its effectiveness remains uncertain.
3. Texture Inconsistencies. Despite color transformation, subtle color and texture shifts remain, questioning the robustness of temporal consistency.
4. Limited Support for High-Dynamic Motion. The model struggles with high-dynamic motion, restricting its applicability in scenarios needing greater flexibility.

**Questions:**

1. Could the authors provide more quantitative insights into the effects of the attention injection mechanism on temporal consistency?
2. How sensitive is the model’s performance to variations in the blending weight (α) during attention injection?

---

> ### Author Response · Authors · 2024-11-22
> **Response to Reviewer gHhE (1/2)**
>
> We thank the reviewer for time and effort in reviewing our paper, making valuable feedback and suggestions regarding the baselines and ablation experiments. Regarding your concerns, we address as follows:
>
> **Q1: Comparison with L4GM & Efficient4D**
>
> - L4GM
>
> L4GM is a concurrent work that uses feed-forward network to directly predict the Gaussian sequences from a monocular video, extending LGM in temporal dimension. Compared to optimization-based method, it shares faster inference speed. However, to train the large-scale 4D model, it needs to collect large scale 4D dataset, and perform extensive training.  And it potentially can not generate high-quality results for out-of-distribution data. In contrast to L4GM, our method generates multi-view video in a training-free manner, then optimizes the 4D representation. Since L4GM has not released the official code yet, we could not provide experimental comparison in the discussion period. Following your suggestions, we have discussed about this work in the related work in **Line 107&131** and discussion on efficiency in **Line 1208-1209**.
>
> - Efficient4D
>
> In the revised paper, we conduct comparative experiment of **Efficient4D**. And we add this comparison in Appendix **Figure 16** and quantitative comparison in **Table 2**. Efficient4D uses temporal synchronization that interpolates over video feature volumes, leading to motion averaging easily. But we use attention injection over multi-view diffusion attention layers. It is evident that Efficient4D produces more blurry results in the side view and in consecutive frames, and we surpass Efficient4D in all quantitative metrics.
>
> **Table R1.1** Quantitative comparison with Efficient4D.
> | Method | CLIP-I | PSNR | SSIM | LPIPS | FVD | CD-FVD |
> | --- | --- | --- | --- | --- | --- | --- |
> | Efficient4D | 0.9358  | 20.38 | 0.822 | 0.179 | 753.11 | 891.33 |
> | Ours | **0.9535**  | **23.28** | **0.904** | **0.173** | **142.34** | **459.10** |
>
> **Q2: Ablation study on attention injection.**
>
> We have compared our attention injection with several variants in **Figure 5 & Line 456-477**. Specifically, we compare the variant on the latent updating method (using EMA blending `S-EMA`/ Linear blending `S-Linear` / residual connections `S-Res`), and which attention layer in SVD to blend (spatial layers `S-EMA` or temporal layers `T-EMA`).
>
> In the revised version **Line 945-956 (Appendix Table 4)**, we update the **quantitative** **ablation** results on different variants of attention injection for temporal consistency and identity preservation.
>
> To quantify temporal consistency, we compute the CLIP-I score between the first frame and subsequent frames. Our results indicate that both `T-EMA` and `S-EMA` significantly improve temporal consistency (inter-frame similarity), achieving higher CLIP-I scores compared to other variants.
> For motion range assessment, we employ the flow intensity, calculated by average value of optical flow on adjacent frames. When CLIP-I score is comparable, larger flow intensity indicates a larger range of motion. The optical flow is estimated with RAFT [1]. ‘-’ means no reasonable results predicted by the optical flow estimator on video with noisy background.
>
> As shown in the following table, we can find that only S-EMA can enhance the temporal consistency meanwhile keeping considerable motion range, demonstrating its clear effectiveness. All blending weights are set to 0.5 for fair comparison.
>
> **Table R1.2**: Quantitative results of different attention injection variant.
>
> |  | CLIP-I  | Flow Intensity |
> | --- | --- | --- |
> | Independent | 0.9323 | - |
> | S-Res | 0.9136 | - |
> | S-Linear | 0.9654 | **2.912** |
> | T-EMA | **0.9962** | 1.102 |
> | S-EMA (Ours) | 0.9925 | 2.756 |

---

> ### Author Response · Authors · 2024-11-22
> **Response to Reviewer gHhE (2/2)**
>
> **Q3: How sensitive is the performance to variations in the blending weight (α)?**
>
> In **Figure 5 left & Appendix Figure 11,** we have ablated on the hyperparameter of blending weight in attention injection qualitatively. Therefore, we state that *“Moreover, we observe that the temporal consistency is highly sensitive to blending weight α”*  in **Line 468-470**.
>
> In the revised paper, we update the quantitative results on hyperparameters of blending weight on overall performance in **Appendix Figure 12 & Line 965-967**.
>
> We find that as the blending weight increasing, the temporal consistency is increased, while the motion range is smaller (indicated by decreasing flow intensity and no-decreasing CD-FVD). The extreme case is that when α=1, the video would be still, simply copying the first frame. Therefore, we select α=0.5, achieving well balance between motion range and temporal consistency, with the best CD-FVD score.
>
> **Table R1.3**: Quantitative sensitivity analysis of blending weight.
>
> | alpha | CLIP-I | CD-FVD | Flow Intensity |
> | --- | --- | --- | --- |
> | 0 | 0.8722 | 1868.12 | 10.321 |
> | 0.1 | 0.8927 | 1579.69 | 7.309 |
> | 0.3 | 0.9315 | 679.16 | 4.971 |
> | 0.5 | 0.9535 | **459.10** | 3.743 |
> | 0.7 | 0.9638 | 568.41 | 1.487 |
> | 0.9 | 0.9699 | 507.12 | 1.021 |
> | 1 | 0.9703 | 504.41 | 1.094 |
>
> **Q4: Temporal inconsistency in 4DGS?**
>
> Although 4DGS shares the fast training/rendering, **empirically** it can easily overfit some artifact brought by the video generation model. It is also evidenced in the concurrent SV4D [2] paper: “*we observe that this dynamic NeRF representation produces better 4D results compared to other representations such as 4D GaussianSplatting, which suffers from flickering artifacts and does not interpolate well across time or views*.”  And in our paper, we have made several efforts to enhance temporal consistency in generated multi-view supervision (attention injection) and 4DGS (affine color transformation and multi-scale training), which have been well ablated in our experiments. Finally, how to mitigate the temporal inconsistency in 4DGS would also be a interesting to explore for general 4D generation/reconstruction community. For example, we could use some regularization terms, such as as-rigid-as-possible (ARAP) loss [3] for Gaussian points constraints.
>
> **Q5: Can not support high-dynamic motion.**
>
> The dynamics of results depends on the video generation model. Some of our results in Figures 3 and 4 showing small motion are actually due to the limitations of SVD, not caused by our framework. As we have stated, '*note that our framework need not be restricted by the SVD necessarily, which can hardly produce large motion*', in **Line 419-420**. Our framework is flexible and can be adapted to other video generation models. To demonstrate this flexibility, we have replaced SVD with AnimateAnyone and present results with larger motion ranges in **Appendix Figure 15**, demonstrating that our framework supports high-dynamic motion.
>
> [1]  Zachary Teed and Jia Deng, RAFT: Recurrent All-Pairs Field Transforms for Optical Flow, ECCV 2020.
>
> [2] Xie et al., SV4D: Dynamic 3D Content Generation with Multi-Frame and Multi-View Consistency, arXiv 2407.17470.
>
> [3] Huang et al., SC-GS: Sparse-Controlled Gaussian Splatting for Editable Dynamic Scenes, CVPR 2024.

---

> > ### Comment · Reviewer_gHhE · 2024-11-25
> >
> > I thank the author for their detailed rebuttal. As most of my concerns have been well addressed, I would like to raise my score to 6.

---

> > > ### Author Response · Authors · 2024-11-26
> > > **Response to Reviewer gHhE**
> > >
> > > Dear Reviewer gHhE,
> > >
> > > We sincerely thank you for your thorough feedback on our paper and the opportunity to improve our work.
> > >
> > > Best regards,
> > >
> > > Paper 5395 authors

---

### Author Response · Authors · 2024-11-22
**General Response**

We thank all reviewers for their time and effort in reviewing our submission and providing constructive feedback. We appreciate their recognition of our motivative pipeline, comprehensive ablation studies, and demonstrated improvements over baselines. Responses to reviewer-specific comments are provided in their respective threads. We have carefully revised the manuscript to improve clarity and include additional results as requested. The revised manuscript includes new experiments and updated descriptions, as detailed below:

**Comparison and discussion with baselines**

- Comparison with Efficient4D qualitatively (**Figure 16**) and quantitatively (**Table 2**);
- Comparison with SV4D qualitatively (**Figure 17**) and quantitatively (**Table 5**);
- Discussion with non-open-sourced baselines (L4GM, Animate3D and Diffusion4D);
- More precise comparison with variant of DreamGaussian4D. (**Figure 3)**.

**Ablative studies**

- Quantitative ablation on variants of attention injection (**Table 4**), and blending weight in attention injection  (**Figure 12**);
- Ablation on video temporal refinement proposed in DG4D for Stage III (**Figure 8-e**);
- Ablation on usage of multi-scale renderer in Stage III (**Table R2.1**).

**More results**

- 4D generation results with more complex image prompt (**Figure 18**).

The revised part is highlighted with blue.

---

### Meta-Review · Area_Chair_rF5N · 2024-12-17

**Metareview:**

This paper receives 3x ratings of 6s. The AC follows the recommendations of the reviewers to accept the paper. The reviewers think that the paper is well-written, the proposed method is effective and the experimental results are good. The weaknesses are well-addressed by the authors during the rebuttal and discussion phases, and the reviewers decided to raise their scores.

**Additional Comments On Reviewer Discussion:**

The weaknesses are well-addressed by the authors during the rebuttal and discussion phases, and the reviewers decided to raise their scores.

---

### Decision · Program_Chairs · 2025-01-22

Accept (Poster)